# An integrative model of cardiometabolic traits identifies two types of metabolic syndrome

Amit Frishberg[1†], Inge van den Munckhof[2†], Rob ter Horst[2], Kiki Schraa[2], Leo AB Joosten[2], Joost HW Rutten[2], Adrian C Iancu[3], Ioana M Dregoesc[3], Bogdan A Tigu[4,5], Mihai G Netea[2,6], Niels P Riksen[2‡*], Irit Gat-Viks[1‡*]

[1]The Shmunis School of Biomedicine and Cancer Research, George S. Wise Faculty of Life Sciences, Tel Aviv University, Tel Aviv, Israel; [2]Department of Internal Medicine, Radboud University Medical Center, Nijmegen, Netherlands; [3]Department of Cardiology, Iuliu Hatieganu University of Medicine and Pharmacy, Cluj-Napoca, Romania; [4]MedFuture Research Center for Advanced Medicine, Iuliu Hatieganu University of Medicine and Pharmacy, Cluj-Napoca, Romania; [5]Babeș-Bolyai University, Department of Biology and Geology, Cluj-Napoca, Romania; [6]Department for Genomics & Immunoregulation, Life and Medical Sciences 12 Institute (LIMES), University of Bonn, Bonn, Germany

**Abstract** Human diseases arise in a complex ecosystem composed of disease mechanisms and the whole-body state. However, the precise nature of the whole-body state and its relations with disease remain obscure. Here we map similarities among clinical parameters in normal physiological settings, including a large collection of metabolic, hemodynamic, and immune parameters, and then use the mapping to dissect phenotypic states. We find that the whole-body state is faithfully represented by a quantitative two-dimensional model. One component of the whole-body state represents 'metabolic syndrome' (MetS) – a conventional way to determine the cardiometabolic state. The second component is decoupled from the classical MetS, suggesting a novel 'non-classical MetS' that is characterized by dozens of parameters, including dysregulated lipoprotein parameters (e.g. low free cholesterol in small high-density lipoproteins) and attenuated cytokine responses of immune cells to ex vivo stimulations. Both components are associated with disease, but differ in their particular associations, thus opening new avenues for improved personalized diagnosis and treatment. These results provide a practical paradigm to describe whole-body states and to dissect complex disease within the ecosystem of the human body.

*For correspondence:
niels.riksen@radboudumc.nl
(NPR);
iritgv@gmail.com (IG-V)

†These authors contributed
equally to this work
‡These authors also contributed
equally to this work

Competing interests: The
authors declare that no
competing interests exist.

Reviewing editor: Edward D
Janus, University of Melbourne,
Australia

## Introduction

In humans, there is a large variability in physiological and molecular parameters, which make up specific baseline phenotypic states. These states differ substantially between individuals, whether measured in healthy individuals or in patients. Even individuals within a well-defined clinical subgroup, such as a group of obese individuals or a group of patients with the same complex disease, demonstrate substantial inter-individual variation in their metabolic, immunological, and physiological phenotypes. Such differences in states often have important clinical implications. For example, specific phenotypic states such as high serum lipids, increased waist circumference, and high blood pressure are associated with a variety of complex diseases and with the severity of disease (*Mørkedal et al., 2011*; *Cameron and Zimmet, 2008*). Heterogeneous phenotypic states can be a barrier to successful therapeutic interventions, since treatments are usually the same for a certain disease ('one-size-

fits-all' treatments) (*Mach et al., 2018*; *Ridker et al., 2012*). Gaining an understanding of phenotypic diversity within relatively homogeneous populations of individuals, and developing frameworks to reliably describe the different phenotypic states, is therefore of fundamental interest and could lead to development of more effective personalized approaches to diagnosis and treatment.

It is generally believed that each individual has a 'whole-body state' in which immune, metabolic, and hemodynamic parameters are coordinately regulated (*Brodin and Davis, 2017*; *Price et al., 2017*; *Trachana et al., 2018*). Current attempts to describe these states typically rely on comparisons between healthy and disease groups (*Bouhaddou et al., 2019*; *Hwang et al., 2019*; *Huang, 2009*). They were thus of limited usefulness to describe the overall phenotypic state in any given individual. For instance, a conventional way to describe the whole-body state in the context of cardiometabolic disease is 'metabolic syndrome' (MetS), which is defined as a small cluster of metabolic and hemodynamic factors that are associated with cardiovascular disease (*Huang, 2009*); as the concept of MetS is focused on cardiometabolic disease, it is not necessarily relevant to other types of disease. Consequently, to date, little is known about the most basic aspects of the whole-body state: its nature and structure in healthy humans, its variation between individuals, and its association with disease have not been well characterized.

Here we provide a framework to describe the whole-body state in individuals, characterize the molecular and functional properties of the whole-body state, and describe the potential applications of the state as a marker of disease. We start with the construction of a comprehensive map of immunological, hemodynamic, and metabolic parameters based on the natural diversity in healthy subjects. The resulting map of parameters (in short, a 'map') is consistent and reproducible in two healthy human cohorts of different age and body mass index (BMI) ranges, and in separate analyses of males and females. We show how prototypical co-regulation patterns in this map are used to construct a robust quantitative model for the whole-body state. The model consists of two components that together represent the wide diversity in healthy subjects – one component ('IM1') explains an average of 25% of the inter-individual variation and the second component ('IM2') explains additional 25% (on average) of the variation. Thus, only two components are sufficient to describe the whole-body state, shedding light on a longstanding goal of describing and understanding phenotypic states. Whereas IM1 resembles the conventional definition of MetS, we highlight IM2 as a nonclassical type of MetS, driven by another set of clinical parameters. Compared to the conventional characterization of the whole-body state by MetS, we observe that the two-component model presents enhanced associations with disease, thus providing an opportunity to improve personalized diagnosis and treatment. We also show that a small number of measured parameters is sufficient for an accurate assessment of the whole-body state, making this a practical model for future applications.

## Results

### Clinical and physiological data in healthy cohorts

In this study, we used two cohorts of healthy individuals with different characteristics and distinct origins. One cohort consists of healthy young adults with BMIs in the normal range (75% aged 18–40 with BMI 18–25 kg/m$^2$; $n$ = 473; data from the 500-FG Project *Ter Horst et al., 2016*). For the other cohort, we sampled older adults who are either overweight or obese (aged 53–78 with BMI >27 kg/m$^2$; $n$ = 126) and are generally healthy (e.g., did not have diabetes and carotid atherosclerosis) (*Ter Horst et al., 2020*). For simplicity, we refer to these two healthy cohorts as 'normal BMI' and 'obesity' cohorts, respectively. Subjects in both cohorts are of Western-European (Dutch) background, and both cohorts are part of the Human Functional Genomics Project (http://www.human-functionalgenomics.org/site/). Characterization of these cohorts can be found in the Materials and methods and supplementary material (*Supplementary file 1A, B* and *Figure 1—figure supplement 1A*).

We performed extensive phenotyping of these individuals with clinical and physiological parameters (in short, 'parameters') relating to four main immunometabolic categories: immune, hemodynamic, metabolism parameters, and lipoprotein parameters (a total of 292 and 259 parameters in the obesity and normal-BMI cohorts, respectively; *Supplementary file 1C, D*). The immunological parameters include the immune composition of blood and the cytokine production capacity of

isolated peripheral blood mononuclear cells (PBMCs) in response to ex vivo stimulation with microbial products. The hemodynamic parameters include cardiovascular parameters such as heart rate, as well as hematological parameters such as hematocrit. The metabolic parameters include metabolites in blood and markers of body fat distribution. Finally, the data consists of 168 lipoprotein parameters, which include the composition of lipids within various subfractions of circulating lipoproteins. The validity of these data sets is demonstrated in *Figure 1—figure supplements 1* and *2*. Throughout this study, we used the relative measured levels of each parameter – that is, for each parameter, its measured levels were centered and scaled across the entire cohort (for simplicity, relative levels are also referred to as 'measured levels').

These two cohorts provide an attractive opportunity for a comprehensive evaluation of reproducibility due to their high overlap of clinical parameters (*Supplementary file 1C*). As more parameters were available for the obesity cohort than for the normal BMI cohort, we first demonstrate our main findings in the obesity cohort and then evaluate the generality of these findings through additional analyses of the normal-BMI cohort.

## An integrative map of clinical and physiological parameters

The map of parameters was constructed in a two-step approach (Materials and methods). In the first step, we applied data transformation in order to amplify signals of similarity among clinical parameters. The transformation is guided by lipoproteins: if two clinical parameters are correlated (after the transformation), it means that they have similar interrelations with lipoproteins. The transformation indeed amplifies the correlations among parameters (*Figure 1—figure supplement 3A–C*). The second step relies on this amplification to define a robust structure of clinical and physiological parameters. Specifically, the structure was inferred from the transformed data using dimensionality reduction (principal component analysis). The result of this two-step procedure is a low-dimensional space (called a 'map') in which each clinical and physiological parameter is embedded in a certain position within the space.

Applying this construction on the obesity cohort yielded a two-dimensional map that could describe a large part of the variability (80%) in clinical and physiological parameters. The two dimensions of the map are referred to as 'axis IM1' and 'axis IM2' (*Figure 1A* and *Supplementary file 2A*). In this map, the various parameters fall in a circular pattern along a nearly continuous spectrum. The circular organization of the map reflects a relatively simple structure of interrelations among parameters: proximal parameters are positively correlated and parameters that are located in opposite sides of the map are negatively correlated (*Figure 1B*, *Figure 1—figure supplement 1B–D,* and *Supplementary file 2B*). For instance, as demonstrated in *Figure 1B*, the proximal glucose and triglycerides parameters are positively correlated (23% correlation of measured levels), and two parameters located in opposing sides of the map – adiponectin and liver fat – are negatively correlated (43% anti-correlation of measured levels). We confirmed that the positions of parameters in the map of the obesity cohort are similar to the positions of parameters in a map that was constructed using the normal-BMI cohort (*Figure 1C*, left), despite major differences in age and BMI ranges (see details of the normal-BMI map in *Figure 1—figure supplement 4A* and *Supplementary file 2A*). In addition, the map is also consistent in separate analyses of males and females (*Figure 1C*, right).

Next, to assess the biological organization of the map, we examined the map within each of the four main immunometabolic categories. We noted a different distribution of each category: lipoproteins span the entire boundary of the circle, hemodynamic parameters are primarily localized at the top-right edge of the circle, metabolic parameters are localized in two defined regions (on the left and top-right sides of the circle), and immunological parameters do not have specific localization (*Figure 1D* and *Figure 1—figure supplement 4B*). We then focused on the organization within the lipoprotein parameters, which consist of multiple measures in 10 lipoprotein subfractions of the 'endogenous' lipoprotein pathway – a central pathway of a gradual change from the subfraction of large very-low-density lipoproteins (XXL-VLDLs) to the subfraction of small low-density lipoproteins (S-LDLs) (*Figure 1E*, right). We found that for each of the particular measures, there is a clear correspondence between the order of subfractions along the known endogenous pathway and the anti-clockwise order of parameters along the map (*Figure 1E* and *Figure 1—figure supplement 4C*), indicating that the circle reflects gradual changes in the maturation of lipoproteins. For instance, all 10 parameters of 'triglyceride content' within 10 different lipoprotein subfractions are ordered along

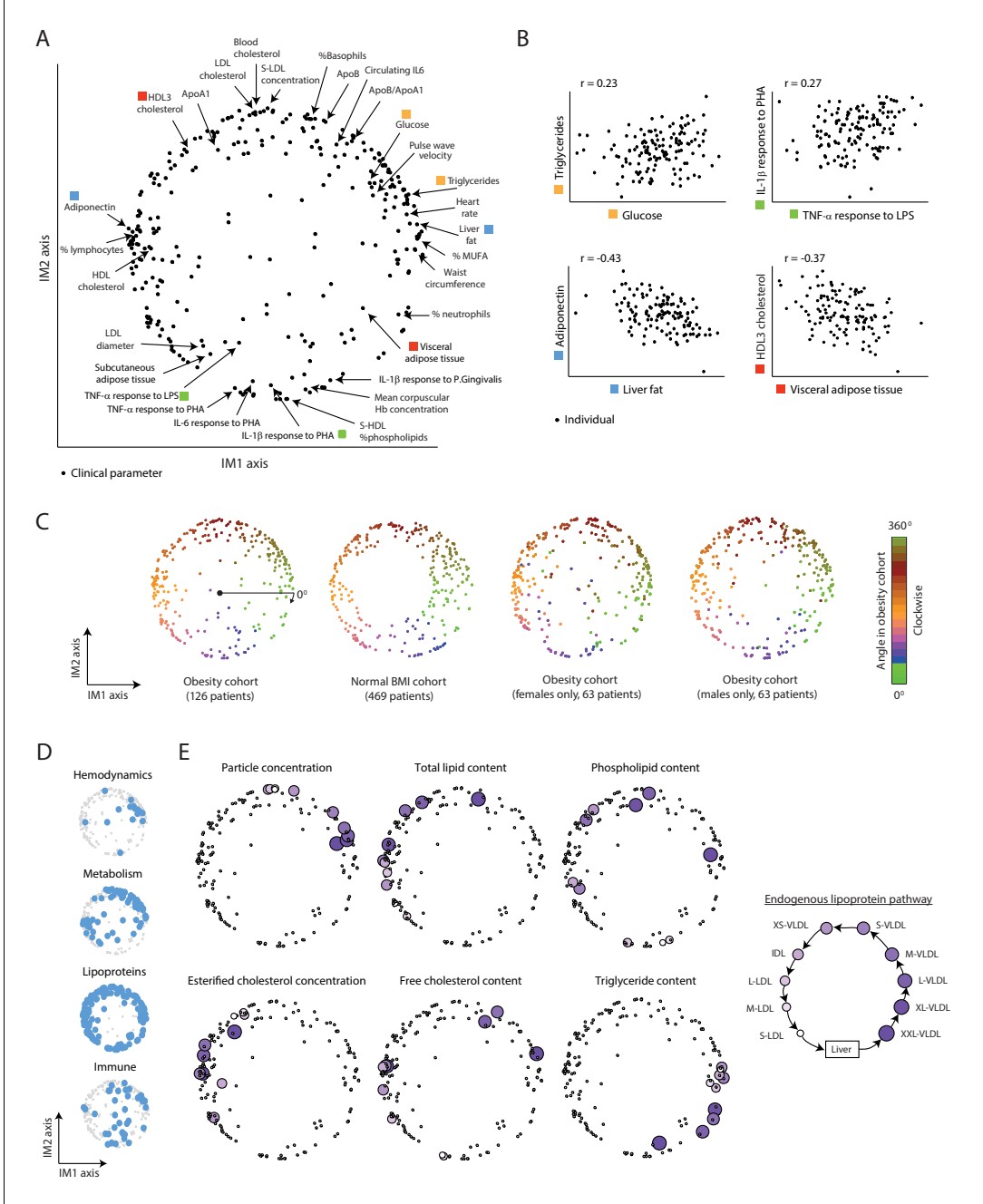

**Figure 1.** A continuous map of clinical and physiological parameters in healthy humans. (**A**) The map of parameters: a map of similarities among clinical and physiological parameters. A principle component analysis (PCA) plot based on dimension reduction in which each clinical and physiological parameter is a dot embedded in a space of two principle components. The principle components are referred to as the IM1 and IM2 axes. Selected pairs of parameters (color coded) are detailed in B. Data are from individuals in the obesity cohort. (**B**) The organization of the map: representative examples. Presented are measured levels of four pairs of clinical parameters (color coded as in **A**). Each individual is represented by a dot. The plots represent the positive correlation between proximal parameters on the map (top) and negative correlation between parameters in opposite sides of the map (bottom). (**C**) Robustness of the parameter map in different healthy subpopulations. Shown are four maps that were constructed independently using the obesity cohort, normal BMI cohort, only females in the obesity cohort, and only males in the obesity cohort (ordered from left to right). All four panels have the same color coding based on the angle of the parameter in the map of the obesity cohort (clockwise color coding starting from right). (**D**) Distribution of biological categories in the map. Shown are the positions of immune, metabolic, hemodynamic, and lipoprotein parameters in the map. (**E**) Lipoprotein data supports the continuous nature of the map. Right: illustration of the known endogenous lipoprotein pathway. The data from the obesity cohort consist of several different measures within each of the 10 lipoprotein subfractions. Left: Each subpanel shows the map from **A** with highlighting the 10 parameters corresponding to one measure that was evaluated in each of the 10 lipoprotein subfractions (colored by

*Figure 1 continued on next page*

*Figure 1 continued*

subfractions, as illustrated on the left). Five of the six measures show gradual anticlockwise changes in positions that are in agreement with the known process of maturation along the endogenous pathway.

The online version of this article includes the following figure supplement(s) for figure 1:

**Figure supplement 1.** Validity of clinical parameters.
**Figure supplement 2.** Analysis of lipoproteins.
**Figure supplement 3.** Relations among clinical parameters.
**Figure supplement 4.** Organization of clinical and physiological parameters in the normal BMI map.
**Figure supplement 5.** Data transformation.

the circular pattern by their known order in the endogenous pathway. Thus, the map is organized by biological properties of the various clinical and physiological parameters.

Collectively, systematic dissection of the natural diversity in healthy subjects generated a comprehensive and reproducible map that can be used as a reference for subsequent analyses of the whole-body state.

## The map enables identification of two main components of the whole-body state – the IM1 and IM2 states

Our next goal was to characterize the overall physiological status of any given individual. To study individual subjects, we started with a personalized visualization of the map of parameters: for each individual, its measurements were presented as color coding of all parameters in the map (indicating the measured levels of all parameters in one individual). Surprisingly, in 91% of the individuals from the two healthy cohorts, we found a global gradient of measurements over the map: in each individual, the measured levels varied from negative (i.e., lower than average) values in one side of the map to positive (i.e., higher than average) values in the opposite side of the map (e.g., obese individuals #1–8 in *Figure 2A* and normal-BMI individuals #9–18 in *Figure 2—figure supplement 1A*). For example, individual #1 has positive measured levels at the top-right side and negative measured levels at the bottom-left side of the map. Individuals are distinctively defined by their spatial direction of their gradients (e.g., eight different directions in individuals #1–8) and by their extent of imbalance between the two sides of the gradient (e.g., strong and moderate imbalanced state in individuals #9–16 versus #17–18, respectively). For 9% of the individuals, there were no major differences in measured levels between the two opposing sides of the map – that is, these individuals are in a 'balanced state' without any observed gradient (e.g., individuals #19 and #20 in *Figure 2—figure supplement 1A*).

Given that most individuals are characterized by a certain global gradient over a two-dimensional map, only two quantitative scores are needed to describe the overall phenotypic state of each individual. In accordance, each individual is described by two scores, called 'IM1 state' and 'IM2 state', which reflect the contributions of the respective axes IM1 and IM2 to the global gradient (Materials and methods). Importantly, the two scores explain substantial fractions of the variation in clinical data, including the variation of each parameter across individuals (the 'percentage of explained inter-individual variation') and among parameters within individuals (the 'percentage of explained inter-parameter variation') (*Figure 2B* and *Figure 2—figure supplement 1B,C*). For instance, in the obesity cohort, for 50% of the individuals, the two scores explain 50% or more of the total inter-parameter variation; similarly, for 50% of the clinical parameters studied, the two scores explain 40% or more of the total inter-individual variation in the obesity cohort. Even when the map from one cohort was used to calculate scores of individuals from the other cohort, the scores still explain substantial percentages of the variation (*Figure 2—figure supplement 1D*). Furthermore, the calculation of IM1 and IM2 states is robust to different sets of parameters (*Figure 2—figure supplement 2*). Thus, these data suggest that the combination of only two numbers (i.e., the IM1 and IM2 states) is sufficient to specify the 'whole-body state' for the clinical parameters included in our cohort data.

As an overview of the whole-body state across individuals, each individual is described by a point whose coordinates are its IM1-state and IM2-state scores (obesity cohort: *Figure 2C*, normal-BMI cohort: *Figure 2—figure supplement 1E*; individuals #1–20 are highlighted). This presentation is

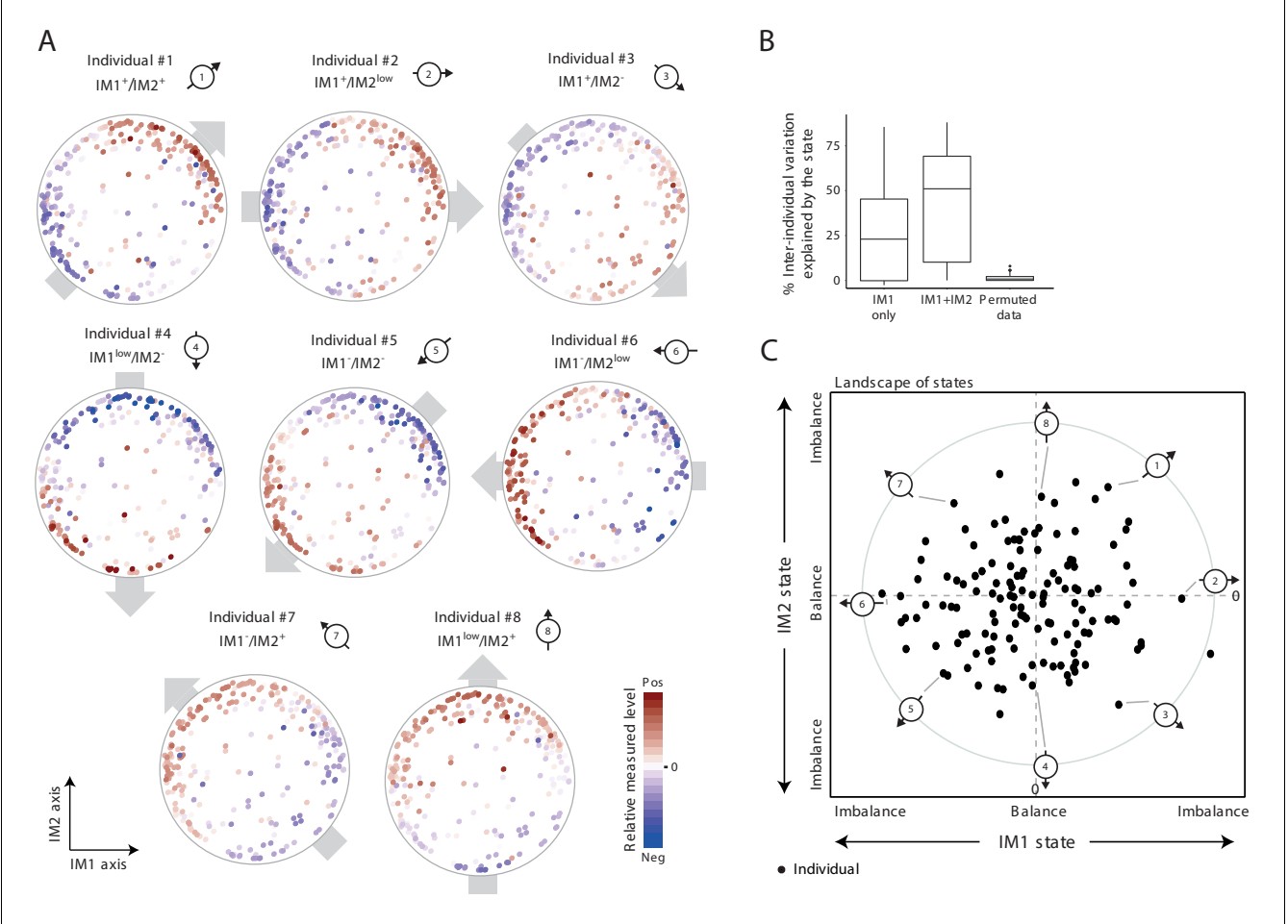

**Figure 2.** The whole-body state consists of two key components – the IM1 and IM2 states. (**A**) Representative patterns of co-regulation over the map of parameters. Shown are measured levels of eight individuals along the map of parameters. Each map (subpanel) is shown with color coding of measured levels from one specific individual: blue/red scale for low/high (relative) measured levels (no data transformation; data were not smoothed). The observed gradient-like patterns of measured levels over the map, indicated with gray arrows, allow reduction of data for each individual into two scores – the 'IM1-state' and 'IM2-state' scores – which together specify the 'whole-body state' of each individual. Original individual identifiers were renumbered as #1–8. For each individual, the signs of the two scores are indicated on top (IM⁻ and IM⁺ for negative and positive scores, IM$^{low}$ for zero and nearly zero scores). (**B**) Box plots of percentage of inter-individual variance that is explained by IM1 state only (left), IM1 and IM2 states (middle) and using permuted data (right). (**C**) The landscape of states. Scatter plot of inferred IM1-state scores (*x* axis) and IM2-state scores (*y* axis) for all individuals in the obesity cohort. The coordinate of each individual (a dot) is a representation of its whole-body state. Indicated individuals (#1–8) are detailed in **A**.

The online version of this article includes the following figure supplement(s) for figure 2:

**Figure supplement 1.** Characterization of the whole-body state in the normal BMI cohort.

**Figure supplement 2.** The IM1-state and IM2-state scores are consistently reproduced with subsets of clinical parameters.

**Figure supplement 3.** Association of parameters with the IM1 and IM2 states.

referred to as a '*landscape of states*'. According to the definition of the IM1 and IM2 states, the direction and distance of an individual from the center of the landscape reflect the direction and extent of its gradient along the map of parameters (Materials and methods). Importantly, the landscape of states indicates that (*i*) there is a similar diversity of IM1 and IM2 states in the healthy population, and (*ii*) the two states are decoupled, with the observation that healthy individuals span the entire spectrum of IM1/2-state combinations. These two observations suggest that the IM1 and IM2 states are of similar importance. Consistent with this notion, the two states indeed have similar contribution to phenotypic diversity: whereas the IM1 state explains 25% (on average) of the variation among obese individuals, the IM2 state explains, on average, additional 25% of this variation

(*Figure 2B*); similarly, the IM1 and IM2 states explain 20% and 28% of the variation among normal-BMI individuals, respectively (*Figure 2—figure supplement 1B*). Particularly, the best explained parameters for each spectrum of states were found at the two extremities of its relevant axis (*Figure 2—figure supplement 3* and *Supplementary file 2A*).

Collectively, we identified two decoupled states (IM1 and IM2) that together describe the human body with respect to a wide array of clinical and physiological measures. The two states have similar contribution to the overall phenotypic diversity.

## Biological characterization of the IM1 and IM2 states

Using the landscape, we were able to demonstrate that distinct states are marked by different clinical characteristics (*Figure 3A*). For example, IM1-positive individuals showed high percentage of monounsaturated fatty acids (MUFAs) and a high waist circumference; IM1-negative individuals showed high high-density lipoprotein (HDL) cholesterol and adiponectin; IM2-positive individuals showed high LDL cholesterol levels and high percentage of basophils; and IM2-negative individuals showed an elevated response of pro-inflammatory cytokines (TNF-α, IL-6, and IL-1β) to bacterial stimulations. Interestingly, IM2 is associated with cytokine responses to all four bacterial stimulations but not to the fungal stimulation (*C. albicans*; *Figure 3B*). Fungal pathogens are mainly recognized by C-type lectin receptors (CLRs), unlike the bacterial stimuli used in the study that are typically recognized by the TLR/RLR pathways (*Plato et al., 2015*), suggesting that pathogen-specific signal transduction may have a role in the creation or maintenance of the IM2-state spectrum. Finally, multiple parameters are markers of a composite IM1/IM2 state – e.g., the ApoB/ApoA ratio is a marker of the IM1-positive and IM2-positive subset of individuals, and triglycerides in L-VLDLs is a marker of individuals carrying the IM1-positive and IM2-negative state (*Figure 3C*). We note that the positions of parameters in the map correspond to their associations with the state, as exemplified in *Figure 3A,C* (see *Figure 2—figure supplement 3* for other parameters). Thus, the map can be used as a bird's-eye view for the particular markers of each state.

Next, to better understand the states, we focused on unique biomarkers for the IM1 and IM2 states (namely, markers of one score that are not associated with the other score; Materials and methods). The top 24 markers primarily consist of lipoprotein parameters (*Figure 3D, E*, *Figure 3—figure supplement 1A* and *Supplementary file 2C*), consistent with the computational construction of the map around lipoproteins (see above for definition of the map). We observed that the IM1 and IM2 markers differ strongly between HDL, LDLs, and VLDLs – particularly, IM1-specific markers consist of VLDLs/intermediate density lipoproteins (IDLs), while IM2-specific markers consist of LDL and HDL subfractions. To understand this observation, we expanded the analysis to all lipoprotein subfractions. We found a continuous change in associations along the lipoprotein pathways – for instance, the IM1 state demonstrates a gradual change in associations with a peak in XS/S-VLDLs (*Figure 3—figure supplement 1B–D*). One exception is the signature of associations with S-HDLs that is different from other HDL subfractions (*Figure 3—figure supplement 1E*).

The identified markers provide important insights into unique functions of each state. (1) The peak of associations with the IM1 state is in XS/S-VLDL parameters (*Figure 3—figure supplement 1B*), which is the last step in the removal of triglycerides from VLDLs to peripheral tissues. Triglycerides are removed from VLDLs into cells in the form of fatty acids and glycerol, which are subsequently used for storage and energy production by beta-oxidation, glycolysis, and gluconeogenesis (*Feingold and Grunfeld, 2018*). Thus, we suspect that the variation in the release of triglycerides from VLDLs, which has an effect on fatty acid metabolism, specifically associated with the IM1 state (*Figure 3F*). This hypothesis is consistent with the observation that IM1 (but not IM2) is linked to the composition of fatty acids in serum (*Figure 3—figure supplement 1F*). Furthermore, expression levels of glycolysis/gluconeogenesis genes and beta-oxidation genes are indeed associated with the IM1 (but not IM2) state across individuals ($p<10^{-6}$, 0.04, respectively, state-function association test, using expression in PBMCs across the normal-BMI cohort; Materials and methods). Collectively, these findings suggest that carbohydrate metabolism in peripheral cells specifically associates with the IM1 spectrum of states. (2) IM2-positive markers include LDL levels and LDL cholesterol, which relate to the influx of cholesterol into peripheral cells and the liver (*Feingold and Grunfeld, 2018*), and IM2-negative markers include phospholipid and free cholesterol in S-HDLs, which relate to the pick-up of free cholesterol together with phospholipid from peripheral cells into S-HDLs (*Feingold and Grunfeld, 2018*; *Figure 3F*). The specificity of this S-HDL association (*Figure 3—*

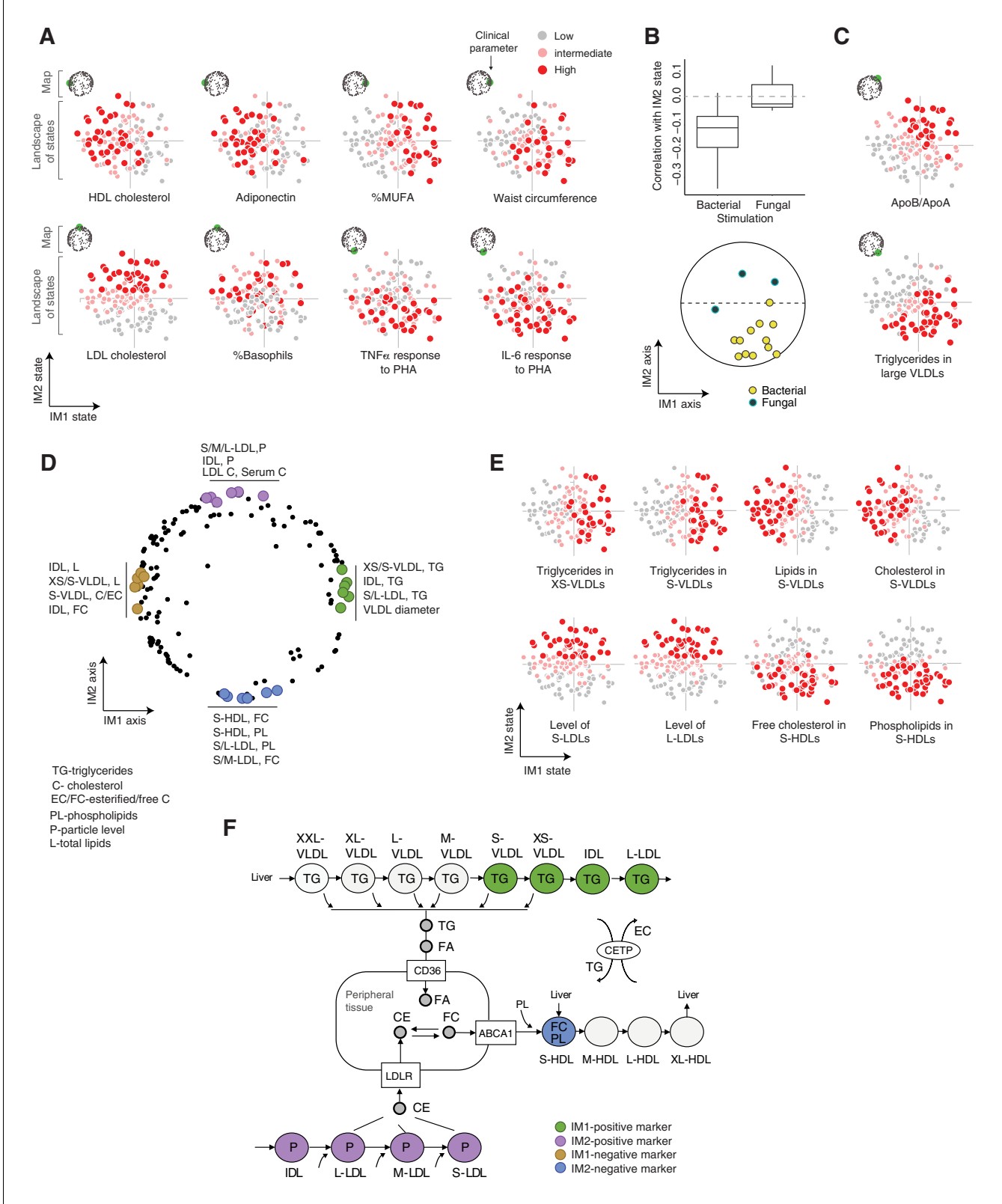

**Figure 3.** Biological characterization of the whole-body state. (**A and C**) Examples of state-specific parameters. For each parameter (a subpanel), shown is the landscape of states (x and y axes are the IM1- and IM2-state scores, respectively), where each individual (a dot) is colored with its measured level of the parameter (gray to red coloring with cutoffs at the 40th/70th percentile). For each parameter, its position within the map is indicated in green (top-left map illustration in each subpanel). **A**, IM1- and IM2-specific markers (q < 0.05). **C**, markers of composite IM1/IM2 states. (**B**) Cytokine response

*Figure 3 continued on next page*

*Figure 3 continued*

of PBMCs is associated with the IM2 state. Top: Distribution of correlations between the IM2 state and the pro-inflammatory response to bacterial (left) and fungal (right) stimulations. Bottom: The position of pro-inflammatory response parameters within the map. Presented are parameters of pro-inflammatory response (TNF-α, IL-6, and IL-1β ) to bacterial (PHA, LPS, *Staphylococcus aureus*, and *Porphyromonas gingivalis*) and fungal (*Candida albicans*) stimulations. (D) Top IM1- and IM2-specific markers, presented by their positions in the map. Highlighted are four groups of six parameters (color coded), a group for each side of each axis. (E) Representative examples of top markers, presented as in A. (F) Marker genes in the context of cellular influx and efflux of lipids. Illustration was compiled from known lipoprotein processes. Markers are color coded as in D. Abbreviations: TG – triglycerides, C – cholesterol, FC – free cholesterol, EC – esterified cholesterol, PL – phospholipids, L – total lipid content, P – particle level.

The online version of this article includes the following figure supplement(s) for figure 3:

**Figure supplement 1.** Top markers for the whole-body state.

**Figure supplement 2.** Associations of biological functions with the state.

**Figure supplement 3.** Estimating the whole-body state from reduced parameter sets.

---

*figure supplement 1E*) is in agreement with phospholipid transfer and the esterification of free cholesterol in subsequent HDL subfractions. Thus, these findings suggest that the spectrum of IM2 states relates to variation in cholesterol homeostasis in peripheral tissues.

To identify the additional functions of the IM1 and IM2 states, we leveraged gene expression data. Specifically, we tested whether a statistically significant fraction of genes in pre-defined functional genesets are associated coordinately with the state (a 'state-function association' test; Methods). We identified 640 functional annotations that are significantly associated (*q*-value <0.05) with the IM1 or IM2 states (*Figure 3—figure supplement 2A* and *Supplementary file 2D*). When top 100 functional annotations were examined, 14% and 25% were broadly categorized as immunological and metabolic functions, respectively. Many of the functions were exclusively associated with one spectrum of state, such as antigen processing, coagulation, proteasome and electron transport chain (ETC) (IM2 positive associations); transcription and chromatin organization (IM2 negative associations); glycolysis, translation, TCA cycle, and DNA replication (IM1 positive associations); and TNF signaling via NFkB (IM1 negative associations). Other functions were significantly associated with both IM1 and IM2, such as RNA processing, transcription factor activity, and interferon/antiviral signaling. IM1 and IM2 are therefore heavily intertwined – e.g., both IM1 and IM2 relate to energy production (glycolysis – IM1; ETC – IM2), immune functions (e.g., TNF signaling – IM1; antigen presentation – IM2), and regulation of proteins (translation – IM1; proteasome – IM2). To validate the associations, we projected gene transcripts onto the map (forming an 'extended map', *Figure 3—figure supplement 2B*, Materials and methods) and then examined areal patterns of functional annotations over the map. Areal patterns were consistent with our predictions: a coordinated associations of genes with the state reflects a pronounce co-localization of these genes in the corresponding region of the map (e.g., *Figure 3—figure supplement 2A,C*). We further note that variation between genes within the same pathway is well fitted to the overall state (e.g., *Figure 3—figure supplement 2D*). Overall, the analysis provides a large repertoire of biological functions that are associated with each spectrum of states.

Taken together, our study suggests that the IM1 and IM2 states are characterized by (1) clinical characteristics – for example, adiponectin and waist circumference (IM1) and pro-inflammatory response (IM2); (2) metabolic processes – for example, cellular metabolism of carbohydrates (IM1) and cellular homeostasis of cholesterol (IM2); and (3) expression of signatures related to molecular functions such as translation, transcription, and RNA processing.

## The state of IM1 is tightly linked to MetS

Given that MetS is an established way to characterize the state of the human body within the context of cardiometabolic risk, we compared the IM1 and IM2 states with the standard MetS classification (using the NCEP ATP-III criteria *Huang, 2009*). We found a strong association between the spectrum of IM1 states and the classification of MetS. First, within the landscape of states, individuals fulfilling the classical criteria for MetS tend toward the positive IM1-state scores ($p<10^{-4}$, *t*-test, *Figure 4A, B* and *Figure 4—figure supplement 1A*). Second, within the map of parameters, all five MetS criteria – low HDL cholesterol, high triglycerides, high waist circumference, high glucose, and hypertension – are nearby the extremes of the IM1 axis (*Figure 4C*); circulating lipoproteins and free cholesterol that are not clustered together with MetS characteristics (*Huang, 2009*) are indeed

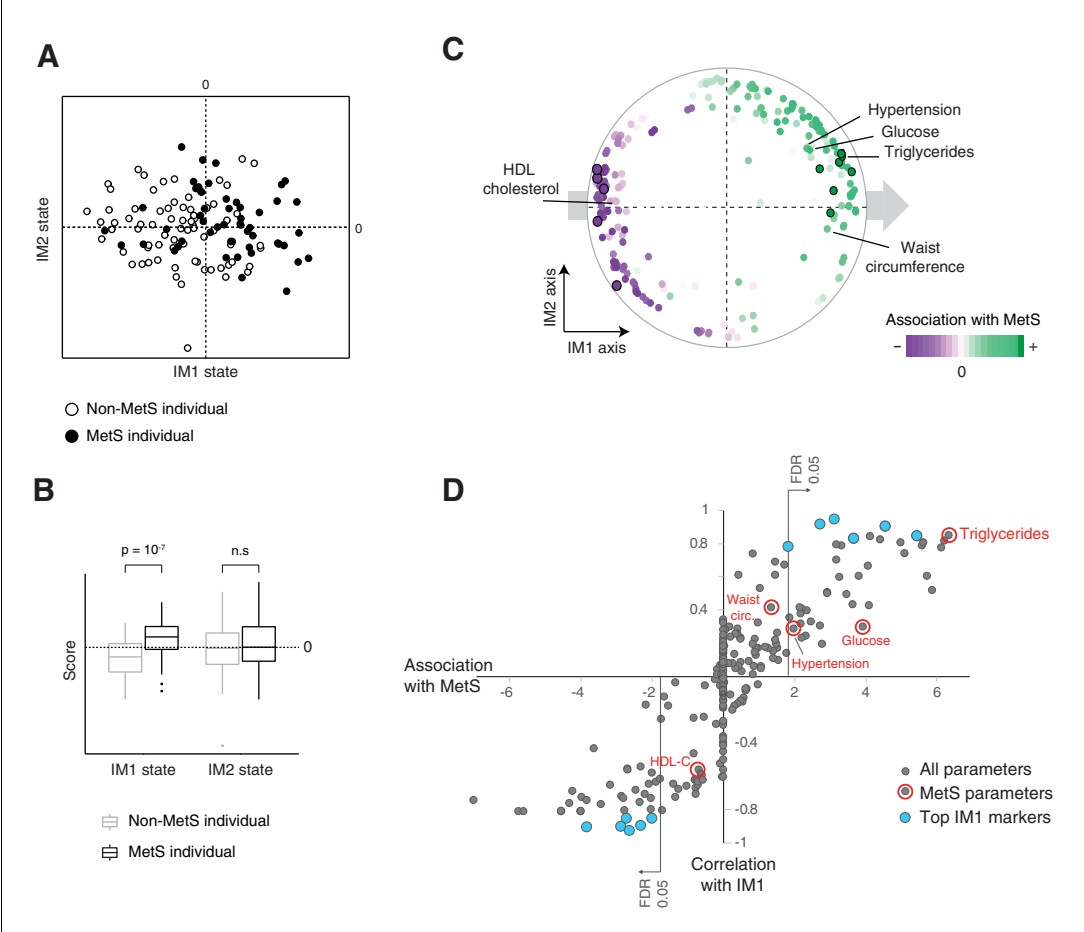

**Figure 4.** The IM1 state is a continuous representation of metabolic syndrome (MetS). (A) The landscape of parameters, color-coded by MetS classification. (B) Box plots of IM1/IM2 states, comparing between MetS and non-MetS individuals. (C) Association of each clinical and physiological parameter with MetS, shown as color coding along the map of parameters. Each parameter (dot) in the map is colored based on its association with MetS (signed −log p-value of association scores, with negative/positive scores in purple/green and outer black circle in top associated parameters; no smoothing). Shown are association scores that account for age and body mass index. The five MetS criteria are highlighted. (D) Comparisons between the association of each clinical parameter with MetS (x axis, calculated as in C) and the association of each clinical parameter with the IM1-state score (Pearson's r; y axis). MetS criteria are highlighted in red. Top IM1-state markers are highlighted in blue. Abbreviations: TG – triglycerides, C – cholesterol, FC – free cholesterol, L – total lipid content, circ. – circumference. MetS classification is based on the National Cholesterol Education Program (NCEP) ATP-III criteria. Results are shown for the obesity cohort.

The online version of this article includes the following figure supplement(s) for figure 4:

**Figure supplement 1.** Relationships between metabolic syndrome (MetS) and the whole-body state model.

located far from the IM1 axis (*Figure 1A*). Third, coloring of the map with the association between MetS and each clinical parameter reveals a global gradient from negative to positive associations along the IM1 axis (*Figure 4C*), which is in agreement with the identified association between MetS and the IM1 state. Fourth, clinical parameters are similarly associated with MetS and the IM1 state (*Figure 4D*). In agreement, the top markers of IM1 (but not IM2) are significantly associated with MetS (*Figure 4D*, *Supplementary file 2C*), the combination of standard MetS parameters predicted the IM1 state with high accuracy (CV $r = 0.87$, $p<10^{-27}$, a linear regression model; *Figure 1—figure supplement 1B,C*), and carbohydrate metabolism has a known role in MetS (*Lonardo et al., 2015*). Finally, analysis of independent data indicates that MetS and the IM1 state share similar associations with disease (diabetes and cardiovascular disease [CVD], see below).

These findings suggest that the state of IM1 is a quantitative representation of the conventional MetS classification. Interestingly, we found that the top biomarkers of IM1 correlate well with MetS (*Figure 4D*), suggesting alternative markers that could be useful in MetS diagnosis. Some IM1-

associated functions have a known role in the pathophysiology of MetS (e.g., S/XS-VLDL-triglycerides *Kolovou et al., 2005*); other IM1-associated functions may offer new insights into the biology of MetS.

Whereas the state of IM1 is linked to MetS, the state of IM2 has substantial contribution to phenotypic diversity but is decoupled from MetS. We therefore reasoned that if the IM2 state is associated with (immunometabolic or cardiometabolic) disease, it is a representation of a novel type of MetS that is uncoupled from the conventional definition of MetS. We therefore next sought to study the whole-body state in the context of disease.

## Analysis of independent disease data identifies IM2 as a novel non-classical type of MetS

We first asked whether patients with specific disease states are represented well by the IM1/IM2-state model that was initially constructed based on healthy subjects. In analysis of two diseases (diabetes mellitus [within the context of obesity] and systemic lupus erythematosus [SLE]; Materials and methods and *Supplementary file 2E*), we observed that the whole-body state of almost all disease patients is indeed represented well by the IM1/IM2-state model – that is, (*i*) disease patients are characterized by a global gradient over the map, (*ii*) there is a wide spectrum of directions and extents of these gradients, and (*iii*) the IM1/IM2 states of disease patients explain much of their observed inter-individual variation (*Figure 5—figure supplement 1A,B*). Thus, the IM1/IM2 states are a general characteristics of the human body in both health and disease. These findings led us to use the same health-based map for the calculation of states in both healthy and disease subjects, providing states that are comparable across different individual groups.

When we compared the states of healthy and disease subjects (*Supplementary file 2E*), we found that both the IM1 and IM2 states are associated with disease, and each disease associates with a different combination of these two states. In particular, diabetes is associated with a positive IM1 state ($p<0.0002$) rather than with an IM2 state ($p>0.05$), in agreement with the fact that MetS is strongly associated with diabetes (*Balkau et al., 2007*). SLE, in contrast, is associated with a positive IM2 state and a negative IM1 state ($p<10^{-10}$, $10^{-8}$, respectively, *Figure 5A,B*). Analyses of additional cohorts show broadly similar behaviors, in both males and females (*Figure 1—figure supplement 1C,D*).

We reasoned that the presence of state-disease associations does not necessarily imply that the entire state of the human body is associated with disease. An alternative possibility is that particular clinical parameters led to the state-disease association because they are associated with disease and are also used for the calculation of the state. Two lines of evidence alleviate this concern. First, similar results were obtained when using a different set of parameters for the calculation of IM1/2-state scores (*Figure 2—figure supplement 2*), suggesting that the associations are not due to several particular parameters. Second, we used two approaches to calculate the association between each clinical parameter and disease: using a standard association test that accounts for variation in gender, age, and BMI (a 'conventional association') and using an extension of the association test that also accounts for variation in IM1/2 states (termed 'state-independent associations'). This analysis confirmed that most associations with clinical parameters reflect the association with the whole-body state: (*i*) although the map of parameters was constructed using only healthy individuals, we observed global gradients over the map for the conventional associations between each clinical parameter and disease (*Figure 5C*, left), which is in agreement with the direction of associations of each disease with the IM1 and IM2 states (*Figure 5A,B*), and (*ii*) accounting for the state drastically reduced the number of significant associations (*Figure 5D*). Similar results were obtained using additional data sets and using an independent map of parameters (*Figure 5—figure supplement 2A,B*). As most parameter-specific associations are largely explained by the global state-disease association, it is unlikely that the association with the state is due to confounding parameter-specific associations.

Because each of the two states is associated with disease but only IM1 associates with the known MetS classification, our data supports the notion that the spectrum of IM2 states relates to a distinct immunometabolic syndrome, hereafter referred to as 'non-classical MetS'. For clarity, the conventional MetS – which relates to the spectrum of IM1 states – is referred to as 'classical MetS' (*Figure 5—figure supplement 3A*).

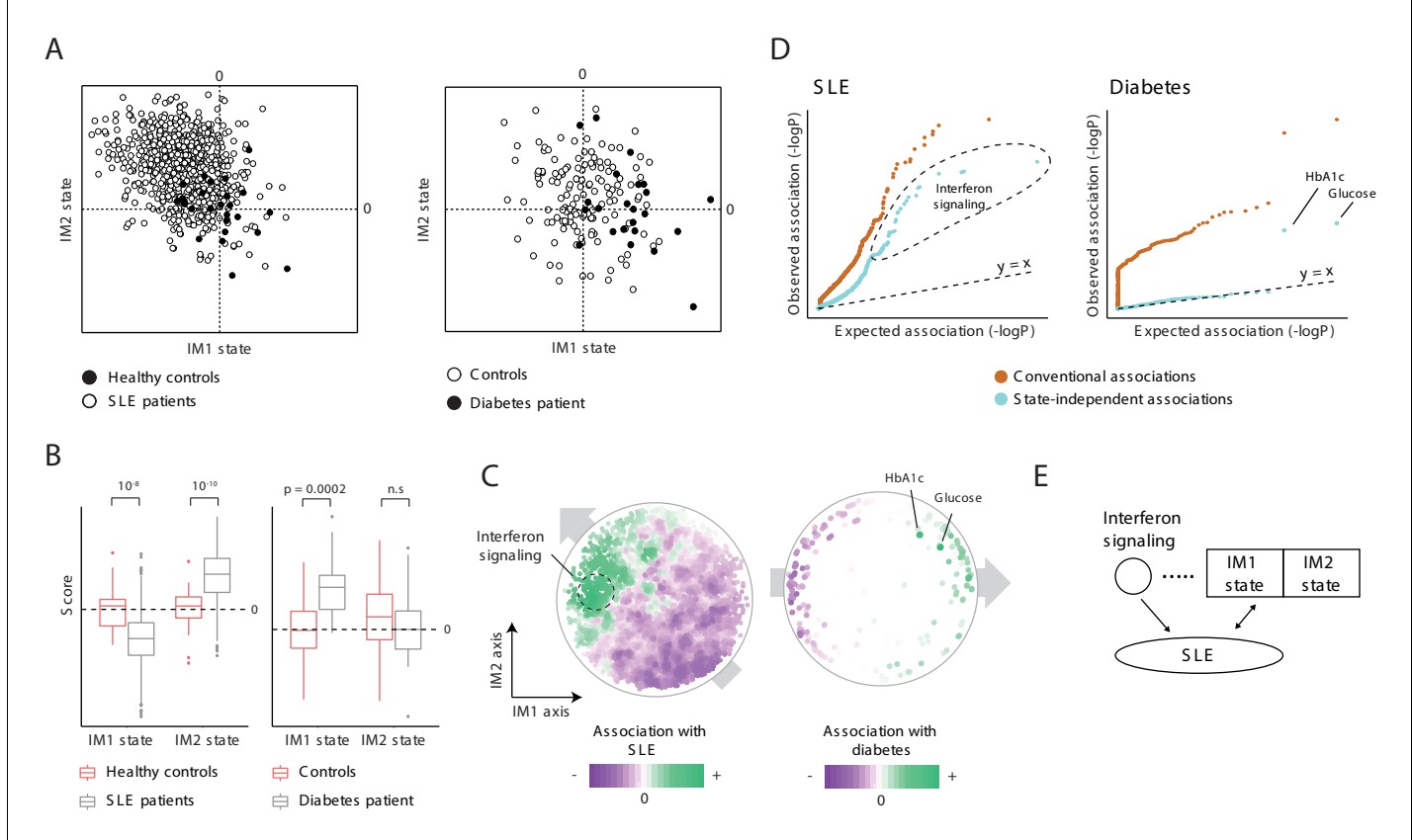

**Figure 5.** Characterization of the IM2 state as a novel immunometabolic syndrome ('non-classical metabolic syndrome'). The analysis is based on independent disease data sets of systemic lupus erythematosus (SLE; left) and diabetes (right) compared to control subjects. (A and B) Disease-specific associations with the whole-body state. The comparison is presented using the landscape of states (A) and as box plots (B), demonstrating bias of each disease toward a certain combination IM1/IM2 states, and emphasizing the utility of IM2 as a novel risk factor of disease. (C) Association of each clinical and physiological parameter with disease along the map of parameters. Each parameter (dot) in the map is colored based on its association with disease (signed −log p-value of association scores, with negative/positive scores in purple/green). Shown are conventional association scores that account for age, gender, and body mass index (BMI). (D) Relations between disease, the state, and specific clinical parameters. A quantile–quantile plot for all tests of association between disease and clinical parameters using conventional association tests (accounting for age, gender, and BMI; orange) and using association tests that account also for the state (light blue). Each plot shows ranked observed associations versus expected associations by an empirical null model; the dashed line (slope = 1) is shown for reference. In both SLE and diabetes, most associations are cancelled out when accounting for variation in states, highlighting a small number of factors that are associated independently of the state. (E) Schematic illustration of the inferred relations between SLE, interferon signaling, and the whole-body state.

The online version of this article includes the following figure supplement(s) for figure 5:

**Figure supplement 1.** Analysis of states in disease patients.

**Figure supplement 2.** Additional analysis of states in disease patients.

**Figure supplement 3.** Relations between disease, specific causal factors, and the whole-body state.

## A unified signature for atherosclerosis

CVD is generally associated with MetS (*Balkau et al., 2007*) and is therefore expected to be linked with a positive IM1 state. We found a role for both IM1 and IM2 in atherosclerosis, and hence in CVD. First, both IM1 and IM2 relate to clinical parameters with a known causal role in atherosclerosis: out of two known causal atherosclerosis factors (*Ference et al., 2017*; *Nordestgaard, 2016*), one is located nearby the most extreme point of the IM1 axis (the percentage of triglyceride in VLDL-remnants [also called 'IDLs']) and one is located nearby the most extreme point of the IM2 axis (LDL cholesterol; *Figure 6A*). More globally, using a literature survey of known CVD risk factors (*Supplementary file 2F*), we found that for both the IM1 and IM2 axes, their positive arms are enriched with previously documented positive CVD risk factor and their negative arms are enriched with documented negative CVD risk factors (*Figure 6A*). As in the case of SLE and diabetes

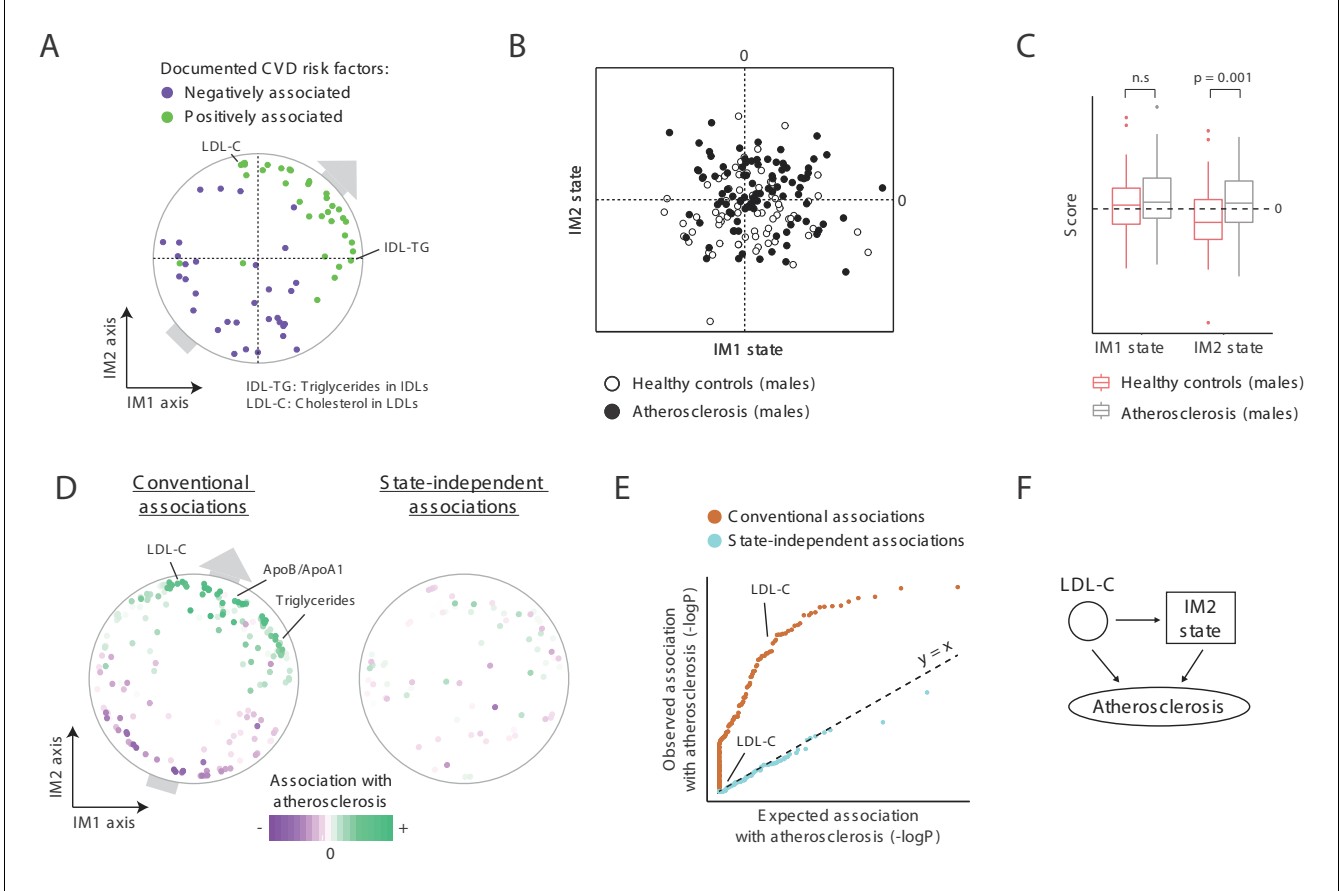

**Figure 6.** A novel signature for atherosclerosis based on non-classical metabolic syndrome. (**A**) Association of the whole-body state with CVD. The map is shown with color coding of parameters by their documented (previously reported) positive or negative association with CVD. Parameters without known associations are not shown. References for these documented associations are listed in **Supplementary file 2F**. The observed trend over the IM1 and IM2 axes implies that both positive IM1 and positive IM2 states are associated with CVD. (**B–E**) Association with atherosclerosis in older adults, obese men, demonstrated using the landscape of state (**B**), box plots (**C**), the map of parameters (**D**), and a Q–Q plot (**E**) (shown as in **Figure 5A–D**, respectively). The plots indicate that the IM2-positive state is associated with atherosclerosis in older adult obese men. (**F**) Schematic illustration for the inferred relations between the IM2-state, low-density lipoprotein(LDL) cholesterol, and atherosclerosis, in obese older adult men.

(**Figure 5C**), such a trend reflects a general association between CVD and positive states of both IM1 and IM2.

Second, focusing on atherosclerosis in obese older adult males, in which reported associations between MetS and atherosclerosis are weak (**Ballantyne et al., 2008**; **He et al., 2007**; **Camhi et al., 2011**; **International Diabetes Federation, 2006**), we indeed found weak relations between positive IM1 states and atherosclerosis (**Figure 6B,C**). Interestingly, in addition to the expected weak association with IM1, we found a significant link between atherosclerosis and an IM2-positive state (p<0.001). These relations are further supported by the trend of parameter-specific associations over the map (**Figure 6D**, left), by the fact that parameter-specific associations are eliminated when accounting for the state (**Figure 6D**, right, and **Figure 6E**), by using an independent map for the calculation of states (**Figure 5—figure supplement 2C**), and by recalculation of association values when excluding diabetic patients (**Figure 5—figure supplement 2D**).

Collectively, these findings suggest that both the positive IM1 state and the positive IM2 state are risk factors for CVD. Thus, the two-component model generalizes the large number of specific risk factors for CVD (**Figure 6A**), offering a unified view of how specific parameters are linked to CVD. We note that the link of atherosclerosis to IM2 is only relevant for males (**Figure 5—figure supplement 1E**), in agreement with the understanding that the mechanisms of atherosclerosis in men are different from those in women (**Fairweather, 2014**; **Mathur et al., 2015**). Thus, whereas both

classical and non-classical MetS are associated with CVD, the relative contribution of each of these syndromes likely depends on the particular context, such as ethnicity, age, gender, and BMI. Overall, our findings coincide with the interpretation of IM1-positive and IM2-positive states as two types of MetS.

## Modeling the relations between the whole-body state, causal factors, and disease

What are the tripartite relations between disease, specific causal mechanisms and the whole-body state? For instance, a causal mechanism and the state could have independent effects on disease, or alternatively, the state could mediate between a causal mechanism and disease. As candidate causal mechanisms, we used clinical parameters that are associated with disease independently of the state (based on the state-independent association test). For each disease, we first describe below the candidate causal mechanisms and then consider the results. Diabetes is omitted because it is only linked to one factor (IM1), as it is associated neither with IM2 nor with any clinical parameter independently of the state (except from its defining elements; *Figure 5D*, right).

To study the relations in SLE, we focused on 88 genes that have significant state-independent associations ($q < 10^{-3}$; *Figure 5D*, left) as the candidate causal mechanism. These genes are primarily responsible for interferon/antiviral signaling ($p<10^{-61}$, hyper-geometric enrichment test), consistent with the known causal role of interferon/antiviral signaling in SLE (*Salloum and Niewold, 2011*; *Supplementary file 2G*). We evaluated a combined multivariate linear model consisting of three factors: IM1, IM2, and averaged expression of the interferon/antiviral-signaling genes. To quantify the relative contribution of each factor, we determined how much additional variance was explained when a given factor was added to the model. Because the different factors may have inter-relations and may interact in a nonlinear manner, all possible ordering combinations were examined. The three factors together accounted for nearly 84% of the variance in SLE. Of these 84%, interferon/antiviral signaling explained 14–80%, and IM1/IM2 explained 4–70%. IM1, IM2, and interferon signaling were independent predictors of SLE ($p<10^{-13}$, $10^{-13}$, and $10^{-103}$, respectively), and each factor was also a significant predictor of disease in a separate analysis of males and females ($p<0.05$ in all cases). Thus, these results suggest that interferon signaling and the classical/non-classical MetS state have independent relationships with SLE (*Figure 5E* and *Figure 5—figure supplement 3B*).

The example of atherosclerosis in obese males is particularly interesting, as in our cohort none of the clinical parameters is associated with atherosclerosis when accounting for variation in the state (*Figure 6E*). Even LDL cholesterol, which has a known causal effect on atherosclerosis (both directly and indirectly *Ference et al., 2017*; *Nordestgaard, 2016*), is strongly associated with atherosclerosis but is completely decoupled from atherosclerosis when accounting for the IM2 state (*Figure 6D, E*). Based on systematic analysis of the tripartite relations between LDL cholesterol, IM2, and atherosclerosis (*Figure 5—figure supplement 3C*), a plausible model is that the indirect effect of LDL cholesterol on atherosclerosis in obese men is mediated through the IM2 state (*Figure 6F*). It is therefore likely that non-classical MetS has an effect on atherosclerosis, in addition to the known effect of specific causal mechanisms.

Overall, these findings suggest that (1) SLE has independent contributions from specific risk mechanisms and the state, and (2) LDL cholesterol affects atherosclerosis both directly and indirectly through the non-classical MetS state. We conclude that human diseases are directly related to the complex ecosystem within the human body, in addition to the contribution of specific disease mechanisms.

## Estimating the whole-body state from reduced parameter sets

We also asked whether the IM1- and IM2-state scores could be predicted well using a small set of parameters, allowing practical assessment of the whole-body state. For the assessment of a given score, we selected arbitrarily four parameters from the 12 top markers (two from each side of the relevant axis, *Figure 3D*) and used a linear regression model to predict the score (Materials and methods). Applying the analysis 100 times, we found that the accuracy is high regardless of the specific four-parameters set used (IM1 mean $r = 0.96$, IM2 mean $r = 0.95$; cross validation Pearson's *r*; *Figure 3—figure supplement 3A*, see example in *Figure 3—figure supplement 3B*). Thus, any small set of top parameters can be used for the assessment of the score. These results

exemplify the power of reduced sets of parameters for the assessment of the whole-body state in practical applications.

## Discussion

In this study, we make use of healthy cohorts in which a vast amount of immunological, metabolic, and hemodynamic parameters were measured, and report that using a tailored transformation of these data, we could construct an informative two-dimensional map of parameters. Using this map, we found that the normal physiological state of the human body is largely characterized by two components – the IM1 state and IM2 state. Although the number of individuals in our data is relatively small, the reproducibility of the map (and thereby the reproducibility of the IM1 and IM2 states) has been validated in three independent cohorts (300-OB, 500-FG, and the REPROGRAM cohorts) and in different individual groups (men and women, health and disease, different age, and BMI groups). A large percentage of the variation in clinical parameters among individuals (50%, on average) is captured by these two components, emphasizing that the two-component model is a faithful representation of the whole-body state. To the best of our knowledge, this provides the first systematic description for the informal concept of the whole-body state and a framework to faithfully describe the state in individuals, shedding light on a longstanding debate in biomedicine.

Studies of cardiometabolic disease have defined MetS as an aggregate of co-regulated clinical parameters that are associated with a high risk of diabetes and CVD (*Balkau et al., 2007*). Our de novo reconstruction of the whole-body state (within healthy subjects) supports this classical definition of MetS: we show that the IM1 component of the whole-body state resembles the classical MetS – that is, the spectrum of IM1 states ranges between negative values in non-MetS individuals and positive values in MetS individuals. Importantly, we further show that the other component of the whole-body state (IM2) reflects a novel 'non-classical MetS': the IM2 spectrum explains substantial fraction of the variation within healthy subjects, is associated with disease risk, and is decoupled from the classical MetS. Thus, our analysis highlights a MetS-independent mode of the human body that is associated with the onset and progression of human disease.

Among our key findings is the identification of the whole-body state as a general risk factor to various complex diseases. Compared to the classical MetS model, the suggested two-component model (*i*) improves disease-state associations and (*ii*) extends the set of diseases that are associated with the whole-body state (*Figure 7A*). This emphasizes the importance of a combined analysis of the two types of MetS: instead of using the classical MetS as a risk factor for a few diseases, different combinations of the classical/non-classical MetS states could be used as risk factors for a larger number of diseases.

An important application of the framework is the ability to discern disease-specific markers whose associations are independent of the whole-body state (e.g., interferon/antiviral signaling in SLE). Thus, our framework offers a strategy for a superior diagnostics based on a combination of general markers (for the state) and specific markers (for a specific disease), which can be tested in future studies. Another application of the model is to study tripartite relations between specific mechanisms, the whole-body state, and disease. For example, we found that SLE has independent contributions from specific disease mechanisms and the whole-body state (*Figure 5E*), and LDL cholesterol affects atherosclerosis both directly and indirectly through the whole-body state (*Figure 6F*). Because it is enough to use a small number of markers for the calculation of the state (*Figure 3—figure supplement 3*), our strategy can be pursued in clinical practice. As our study cohorts are limited in size and are focused on specific genetic background/ethnicity, future work is needed to test the generality and repeatability of the identified associations and relations in additional populations and contexts.

The analysis specifies a large set of factors that are associated with non-classical MetS. These include a mixture of metabolic factors (e.g., serum cholesterol, LDL cholesterol, LDL particle number, and the composition of phospholipid/free-cholesterol in LDLs and S-HDLs), immune factors (e.g., the percentage of circulating basophils and cytokine production capacity of PBMCs to bacterial stimuli), and transcriptional regulation of intracellular functions (e.g., coagulation, antigen processing, and presentation). By leveraging the fine granularity of lipoproteins, the analysis provides specific hypotheses for the roles of cholesterol metabolism in non-classical MetS. We do not know the

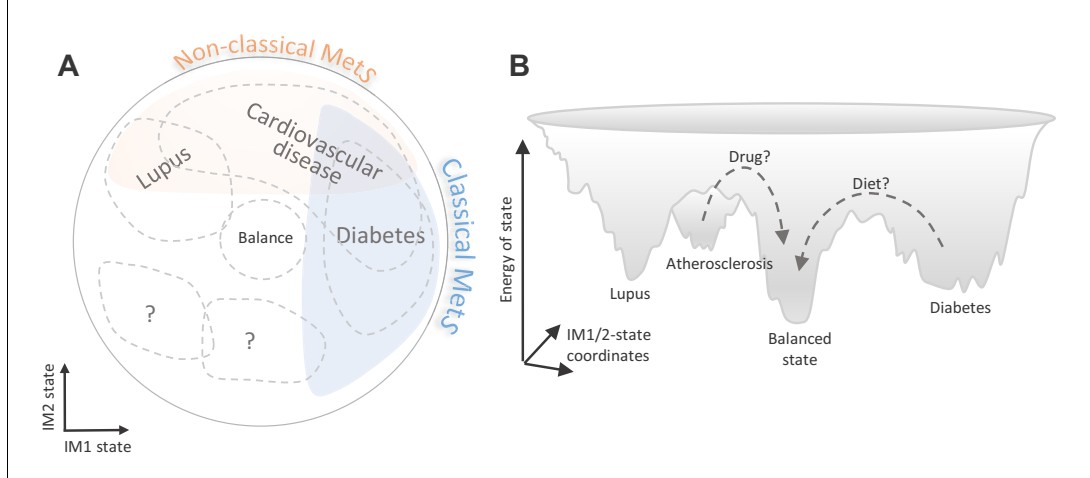

**Figure 7.** Implications of the whole-body state for disease research. Shown are two schematic representations for the landscape of states, highlighting a potential future utility of these states in diagnosis and treatment. (**A**) Distribution of disease risk over the landscape of states. Indicated are regions in the landscape that have a higher disease risk (gray outline). The conventional ('classical') metabolic syndrome (MetS) and the newly defined 'non-classical MetS' are indicated as blue and orange regions, respectively. The landscape highlights the utility of classical/non-classical MetS scores as risk factors for common, complex disease. (**B**) Energy landscape of the whole-body state. Each state in the landscape is associated with a certain level of energy. Possible sites of disease risk are also indicated. The energy landscape highlights our hypothesis that transitions from one state to another state are blocked by energetic boundaries. Thus, transitions from a disease-associated state to a low-risk state may therefore require a certain activation energy, such as changes in lifestyle and drug treatments (dashed lines).

identity of the true drivers of non-classical MetS, as the relations between these parameters consist of multiple intertwined feedback loops.

Our unbiased approach offers a comprehensive view of how biological functions are linked to MetS. The IM1 state associates with the standard MetS parameters and with additional parameters (e.g., adiponectin, %MUFA and the composition of XS/S-VLDLs, cellular metabolism of carbohydrates, protein translation, DNA replication, and TNF signaling). These include many of the known MetS-associated functions (e.g., fatty acid metabolism; *Lonardo et al., 2015*), as well as previously unknown candidates. Thus, the integrated analysis allows to identify new candidates even in the heavily investigated MetS state. Given that the IM1 markers are more strongly associated with IM1 compared to the five conventional MetS parameters (*Figure 4D*), we hypothesize that the IM1 markers may describe in more detail the (patho)physiological MetS state of individuals. It is therefore possible that the definition of MetS might be improved with IM1-specific markers, potentially leading to a stronger association of classical MetS with cardiometabolic disease. Additional studies are required to test this hypothesis and explore the association of IM1 markers with cardiometabolic disease across large cohorts.

Through inspection of the states, we found that most individuals exist in a certain imbalanced state in which one side of the map is upregulated whereas the opposite side of the map is downregulated. Cellular immunometabolism is indeed thought to be in a 'dynamic balance' between antagonistic programs, such as a balance between the pro- and anti-inflammatory signaling and a balance between cell proliferation and apoptosis (*Cicchese et al., 2018*; *Guo and Hay, 1999*). Our observations suggest that similar rules of homeostatic balance are prevalent not only at the molecular and cellular level, but are also effective at the clinical and physiological scale.

Finally, our framework offers opportunities for future medical interventions targeting the state. In particular, the ability to quantify the relations between different diseases and the whole-body state provides a unique opportunity to design changes in the whole-body state (*Figure 7B*). For example, an attractive possibility is to use the landscape of states to design transitions from a high-risk state into a low-risk state. Given that some transitions are likely blocked by energetic boundaries, activation energy (such as drug interventions or long-term changes in lifestyle) would be required to go through these boundaries. We therefore expect that the landscape of states would have an impact on the development of future therapeutic interventions.

## Materials and methods

### The 300-OB cohort

The 300-OB cohort, which we generated as a part of the human functional genomics project (HFGP), consists of 167 males and 135 females (*Supplementary file 1A*). These Dutch volunteers are all over-weight or obese (BMI >27) and range in age from 53 to 79 years. The presence of diabetes and carotid atherosclerosis were determined. In particular, the presence of atherosclerotic plaques was determined by ultrasound of the carotid artery. Subjects with a recent cardiovascular event (myocardial infarction, transient ischemic attack, or stroke in the previous 6 months), a history of bariatric surgery or bowel resection, inflammatory bowel disease, renal dysfunction, increased bleeding tendency, use of oral or subcutaneous anti-coagulant therapy, or use of thrombocyte aggregation inhibitors other than acetylsalicylic acid and carbasalate calcium were excluded. The study was approved by the Ethical Committee of the Radboud University Medical Center Nijmegen (NL46846.091.13, 2013/505). Inclusion of volunteers and experiments were conducted according to Declaration of Helsinki, and all subjects signed written informed consent.

### Experimental procedures for the 300-OB cohort

We took blood samples in the morning following an overnight fast. Participants who used lipid-lowering therapy temporarily discontinued this medication 4 weeks prior to the measurements. For each of the individuals, a variety of immune, metabolic, and hemodynamic parameters were monitored (*Supplementary file 1C,D*). We used a high-throughput nuclear magnetic resonance (NMR) metabolomics platform (Nightingale's Biomarker Analysis Platform) for the quantification of lipid and metabolite measures. Circulating inflammatory mediators were measured in human EDTA plasma using Enzyme Linked Immunosorbent Assay (ELISA). All individuals from the 300-OB study underwent a cardiovascular assessment. This included the measurement of carotid intima-media thickness (cIMT), plaque presence, and maximum plaque thickness. After a resting period of at least 30 min, baseline resting diameter and wall thickness of both carotid arteries were assessed by a well-trained sonographer. The cIMT and diameter measurements were performed in the proximal 1 cm straight portion of the carotid artery at three different angles (90°, 120°, and 180°) for six heart beats. The primary outcome variable was defined as the mean cIMT of the three angles. The presence of plaque was defined as focal thickening of the wall of at least 1.5× the mean cIMT or a cIMT >1.5 mm according to the Mannheim intima-media thickness consensus.

### Healthy cohorts

#### The obesity cohort

The 300-OB cohort includes 126 healthy volunteers (63 males and 63 females without atherosclerosis and diabetes), all are older adults (range from 53 to 78 years) who are either overweight or obese (BMI >27). This healthy cohort of 126 individuals is referred to as the 'obesity cohort' (*Supplementary file 1A*). Volunteers and experimental procedures that we used to generate the cohort are described in detail in 'The 300-OB cohort' section.

#### The normal BMI cohort

We further used the 500-FG cohort (*Ter Horst et al., 2016*), which includes 473 healthy individuals (202 males and 271 females) of Caucasian origin. The individuals range from 18 to 75 years of age with 82% of individuals younger than 30. Individuals had BMIs ranging from 15 to 35 with the majority (388 individuals) in the normal BMI range (18–25). None of these individuals were diagnosed with atherosclerosis or diabetes. We refer to this cohort as the 'normal BMI cohort' (*Supplementary file 1B*).

#### Clinical parameters

The parameters of the obesity and normal-BMI cohorts relate to three main immunometabolic categories (*Supplementary file 1C,D*). The first category consists of immunological parameters including the immune composition of blood (e.g., cell quantities and circulatory levels of cytokines and adipokines; 19 and 8 parameters in the obesity and normal BMI cohorts, respectively). In addition, it includes cytokine production capacity of isolated PBMCs in response to ex vivo stimulation with

microbial products. For example, in the obesity cohort, data includes levels of four cytokines (IL-1β, IL-1RA, IL-6, and TNF-a) in response to five stimulations (phytohaemagglutinin [PHA], lipopolysaccharide [LPS], *Staphylococcus aureus*, *Porphyromonas gingivalis*, and *Candida albicans*). The second category of clinical measurements is a collection of hemodynamic parameters, including heart rate, pulse wave velocity, and blood pressure, as well as hematological parameters such as hematocrit. These hemodynamic parameters were only measured in the obesity cohort (15 parameters). The third category consists of metabolism-related parameters: concentrations of fatty acids, sugars, amino acids, lipids, and apolipoproteins in blood; insulin resistance index; and markers of body fat distribution such as visceral and subcutaneous abdominal fat (71 and 59 parameters in the obesity and normal BMI cohorts, respectively). This category also consists of a high-resolution map of lipoproteins, consisting of 10 subfractions over the VLDL to LDL 'endogenous' pathway and four parameters over the HDL 'reverse transport' pathway, with 12 compiled parameters for each subfraction – including the content of phospholipids, triglycerides, esterified cholesterol, and free cholesterol (a total of 168 lipoprotein parameters for each cohort). Importantly, MetS was assessed only in the obesity cohort but not in the normal-BMI cohort.

## Data pre-processing

For each parameter, centering and scaling were done by fitting all measured levels across a given cohort to the Johnson function. The resulting 'relative measured levels' (in short, 'measured levels') were used throughout this study. The data set of all relative measured levels of all clinical parameter is referred to as 'measured data'. Given the limited number of missing values (an average of 5% missing values across parameters), we performed all analyses without any data imputations. In all individual-level analyses, parameters with more than 65% missing values were omitted. For 68 individuals from the normal BMI cohort, RNA-seq data of PBMCs were available from *Bakker et al., 2018* Raw reads of RNA-sequencing were initially normalized across the samples, and genes were centered and scaled by fitting to the Johnson function. We identified 11,810 genes that were expressed in at least 63 of the 68 individuals and used them to reconstruct the extended map of parameters.

## The map of parameters

To generate the map of parameters, we first transformed the data so as to reduce the level of noise while maintaining the original interrelations among clinical parameters. To this end, we tested the correlation between each pair of parameters across individuals. As many parameters lacked meaningful correlations with any other parameter and the lipoproteins showed exceptionally strong correlations (*Figure 1—figure supplement 5A*), we transformed the measured levels of each clinical and physiological parameter based on its collection of associations with the entire set of lipoprotein parameters. Thus, the transformation represents the measured data of each clinical parameter as a large set of associations with all lipoprotein parameters – substantially different from the original representation in which each clinical parameter is described by a set of measured levels in all individuals. This transformation is based on the notion that (i) associations would allow amplification of correlations ('similarities') among clinical parameters, since they provide a reduced level of noise: small fluctuations in measurements of specific individuals are smoothly averaged at the association level; and (ii) the associations maintain the original interrelationships among clinical parameters: if two clinical parameters behave in a similar way, the complete sets of associations with lipoprotein parameters would be potentially similar.

We tested two alternative transformation methods – a linear transformation that does not account for covariates and a covariate-corrected transformation that accounts for variation in age, gender, and BMI. In both cases, the input is a 'measured profile' of each clinical parameter (represented as an *n*-length vector of relative measured levels across the *n* individuals), and the output is a 'transformed profile' for each clinical parameter.

I. *A linear lipoprotein-based transformation.* We used the lipoprotein data as a transformation matrix that sends each clinical parameter from a profile of measurements across individuals to a new profile of relations (cosine of angles) between a given parameter and all lipoprotein parameters. In particular, for a given clinical parameter *k*, we applied the following linear transformation: $T_k = X \cdot Y_k$, where $Y_k$ is the measured profile of the *k*-th parameter (using

unit vectors for all parameters), $X$ is the $168 \times n$ transformation matrix in which entry $X_{ij}$ is the measured profile of the $i$-th lipoprotein parameter ($i$=1,...,168) in individual $j$ ($j$=1,...,$n$), and finally $T_k$ is a transformed 168-length vector of the $k$-th parameter, referred to as the 'transformed profile'. Since the measured profiles are unit vectors, the $i$-th position in the transformed profile of parameter $k$ ($T_{k,i}$) reflects the cosine of angle between the $n$-length vectors of relative measured levels for the $k$-th clinical parameter and the $i$-th lipoprotein parameter. Thus, the transformed profile of any given parameter $k$ is a vector of all its relations with the list of lipoprotein parameters.

II. *A covariate-corrected lipoprotein-based transformation.* We reasoned that the linear transformation defines relationships of any parameter to the lipoprotein parameters, and that these relationships are possibly biased due to certain covariates such as age, gender, and BMI. To account for these potential confounding effects, we used an alternative transformed profile $T_k$ in which $T_{k,i}$ reflects the relations between clinical parameter $k$ and lipoprotein parameter $i$ while accounting for covariates. In particular, for a given clinical parameter $k$ and a lipoprotein parameter $i$, we used a standard linear regression model $Y_k = c_0^{k,i} + c_1^{k,i} X_i + c_2^{k,i} A + c_3^{k,i} G + c_4^{k,i} B$, where $Y_k$, $X_i$, $A$, $G$, and $B$ represent $n$-length vectors of data across the $n$ individuals including the measured profiles (unit vectors) for clinical parameter $k$ ($Y_k$) and lipoprotein parameter $i$ ($X_i$), as well as age ($A$), gender ($G$), and BMI ($B$). Using this formulation, we set $T_{k,i} = c_1^{k,i}$. Thus, $T_{k,i}$ reflects the relations between the lipoprotein parameter $i$ and the clinical parameter $k$ while accounting for covariates.

Transformations based on associations with other sets of parameters (instead of lipoproteins) were not as effective as a lipoprotein-based transformation (*Figure 1—figure supplement 5B*). As both the linear transformation and the covariate-corrected transformations provided qualitatively similar results (*Figure 1—figure supplement 5C*), in this study we focus only on maps that were generated using the covariate-corrected transformed profiles.

Using data from a healthy cohort, the map of parameters was constructed in two steps. First, measured levels of all clinical and physiological parameters were transformed using the covariate-corrected lipoprotein-based transformation. Transformed values were then scaled so that the transformed profile of each clinical parameter had unit length. Second, the map of parameters was constructed by dimension reduction (PCA) of this transformed-and-scaled data. The two first principle components are termed 'immunometabolic axis 1' (in short, 'IM1 axis') and 'immunometabolic axis 1' (in short, 'IM2 axis').

## The extended map of parameter

To extend the map with gene transcripts, we focused on transcription profiles of PBMCs that were measured in 68 individuals from the normal-BMI cohort, which enabled us to project each gene onto the map of parameters. This extension was applied in two steps: First, we constructed the map using clinical and physiological parameters based on data from the normal-BMI cohort. Second, each gene parameter was projected onto the resulting PCA map (a total of 11,810 genes; see 'Data preprocessing' section). The result is an 'extended map' consisting of 11,810 new parameters where each new parameter is an expressed gene (*Figure 3—figure supplement 2B*). The extended map was used in the analysis of biological annotations and in all analyses of SLE.

## The whole-body state model

The calculation of the whole-body state for each individual $i$ takes as input (i) a map of parameters, where each parameter is associated with a certain coordinate on the IM1 and IM2 axes, and (ii) the relative measured levels of each parameter for individual $i$. The entire calculation of states relies on the relative measured levels (namely, without using the lipoprotein-based transformation).

To infer the state of individual $i$, we solved the following linear regression: $Z_i = s_{IM1}^i V_1 + s_{IM2}^i V_2$, where $Z_i$ is the vector of all relative measured levels of individual $i$, $V_1$ and $V_2$ are the vectors of coordinates of all parameters on the IM1 and IM2 axes, respectively (namely, the coordinates of all clinical parameters in the map of parameters), and $s_{IM1}^i$ and $s_{IM2}^i$ indicate the unknown 'IM1 state' and 'IM2 state' (respectively) in individual $i$. The IM1/2-state scores are signed based on the direction of the gradient, with a positive IM1 state (IM2 state) for upregulation in the right (top) side of the map and vice versa. For instance, the scores of an individual with downregulation in the top-left region of the map and upregulation in the bottom-right region of the map are IM1-positive and IM2-negative

(e.g., individual no. #3 in *Figure 2*). The distance between the $(s^i_{IM1}, s^i_{IM2})$ coordinate and the (0,0) coordinate reflects the 'extent' of regulation along the gradient: the higher the distance, the stronger the upregulation and downregulation along the gradient. The two scores equal zero (or nearly zero) for individuals that lack any global gradient (e.g., individuals #19 and #20 in *Figure 2—figure supplement 1A*).

Overall, the two scores define the 'whole-body state' of each individual. If one or two of the scores differ from zero, there is an observed global gradient of relative measured levels over the map, as if there is an imbalance between two opposing sides of the map. In contrast, if both the IM1-state and IM2-state scores are zero (or nearly zero), there is a lack of global gradient, that is, there is a balance between opposing sides of the map (see indication of imbalanced and balanced states in *Figure 2C* and *Figure 2—figure supplement 1E*). Of note, the calculation of states in disease subjects should also consider potential artifacts due to disease-specific alterations, as detailed in the 'Application of the framework in disease research' section.

For the analysis of SLE, where clinical parameters were not available, the IM1/IM2-state scores were calculated using transcriptome data, relying on an extended map of gene transcripts. Particularly, we used the extended map that was constructed based on data from the normal-BMI cohort (i. e., the map from *Figure 3—figure supplement 2B*). Out of 11,810 genes in this extended map, 7443 were also expressed in the SLE data sets (see 'Disease cohorts' section). To calculate the states, we used only 596 of the 7443 genes, which were selected based on their relations with lipoproteins in two steps. First, for each gene, we calculated its Pearson correlation with all lipoprotein parameters based on gene expression and lipoprotein data from the normal-BMI cohort (correlations were calculated using transformed profiles). Second, genes with maximal absolute-correlation values that are higher than 0.85 were selected.

We present the calculated IM1/IM2-state scores across a cohort of individuals using a 'landscape of state' – a scatter plot in which each individual is described as a point whose coordinates are its IM1-state and IM2-state scores. In this presentation, the gradient of each individual over the map of parameters is specified by its particular position within this landscape: the direction of a coordinate (relative to the center of the landscape) equals the direction of the gradient along the map of parameters, and the distance of a coordinate from the center of the landscape reflects an increasing extent of imbalance between the two sides of the map. Specifically, individuals located at the center of the landscape lack any gradient (i.e., in a balanced state). *Figure 2A,C* and *Figure 2—figure supplement 1A,E* provide several examples. For instance, individual #1, whose gradient is directed toward the top-right side of the parameter map, attained IM1-positive and IM2-positive scores; in accordance, it is embedded at the top-right side of the landscape of states. Individuals #17 and #18 that have a moderate imbalance are located closer to the center compared to individuals #1–16 that have stronger imbalance. Individuals #19 and #20, which lack any global gradient, reside at the center of the landscape.

## Analysis of variation in the whole-body state

For each individual, the 'total inter-parameter variation' is defined as the variation in the relative measured values across all parameters. For each parameter, the 'total inter-individual variation' is defined as the variation in the (relative) measured levels across individuals (here, this total variation is always one due to the scaling of measurements, as detailed above). For both the inter-parameter variation and inter-individual variation, we calculated the percentages of variation that are explained the IM1-state and IM2-state scores in two steps: First, for each parameter, we regressed out the two scores from the parameter's relative measured levels across individuals. Second, the 'unexplained variation' is the variation calculated when using the resulting residual values (rather than using the relative measured levels). The 'percentage of unexplained variation' is the percentage of unexplained from the total variation, and the 'percentage of explained variation' (either inter-individual or inter-parameter) is the remaining percentage of variation – that is, 100 minus the percentage of unexplained variation. This analysis is reported in *Figure 2B, Figure 2—figure supplement 1B–D*, and *Figure 5—figure supplement 1B*.

## Biomarkers and functional annotation of states

Biomarkers for the states were identified based on Pearson's correlation ($r$) between state scores and measurements. The top biomarkers were selected in three steps. First, correlations of lipoprotein content and percentage (for the same lipid and subfraction) were averaged. Second, we averaged the correlations calculated in the obesity and normal-BMI cohorts. Third, we identified the top markers. For example, for the identification of IM1-specific markers, we filtered out all markers with high (>0.3) averaged correlation to IM2 and then selected the top parameters based on their averaged correlation with IM1 (six best positively correlated and six best negatively correlated with the IM1 state). The same cutoffs were used for the identification of IM2-specific markers. *Supplementary file 2C* provides the list of the identified markers.

We predicted the state using a small set of parameters. Specifically, we predicted the two scores as two independent tasks, allowing reduced costs when only one score is needed. For the assessment of a given score (IM1 state or IM2 state), we chose arbitrarily two negative parameters and two positive parameters (four parameters in total per score) from the set of 12 top markers (*Supplementary file 2C*). We fitted a linear regression model that used the score as the dependent variable and the four selected parameters as the explanatory variables. Prediction accuracy (Pearson's $r$) was evaluated using leave-one-out cross validation.

To explore the relations of biological functions with the state, we correlated the expression levels of each gene with each state across individuals (Pearson's $r$ score). Next, for each state (IM1 or IM2 states), gene set enrichment analysis (GSEA) was applied (using the 'fgsea' R package) to test the similarity between the expected and observed distribution of Pearson's $r$ values (comparing between the gene set to all remaining genes). p-values were assessed using permutations. Significant p-values indicate state-function associations. Candidate gene sets of biological functions were downloaded from the following sources: GO, MSigDB Hallmark gene sets, Reactom, and KEGG. Significant function-state associations ($q$-value <0.05) are reported in *Supplementary file 2D* and *Figure 3—figure supplement 2A*.

## Disease cohorts

Each analysis relies on three input types: a group of disease patients, the group of matching controls from the same cohort data, and a reference map of parameters.

### Systemic lupus erythematosus

Three comparisons of diseased to healthy groups were performed: (1) 26 healthy and 719 SLE female children from Banchereau et al. (GSE65391), (2) 6 healthy and 87 SLE male children from Banchereau et al. (GSE65391), and (3) 17 healthy and 131 SLE female adults from Chiche et al. (GSE49454). Since SLE cohort no. 1 is the largest, we used it as our primary analysis (*Figure 5*), and the remaining cohorts are used as additional support (*Figure 5—figure supplement 1D* and *Figure 5—figure supplement 2A*). These three data sets consist of gene expression profiles, and in accordance, the analysis was performed in the context of the extended map of genes from *Figure 3—figure supplement 2B*.

### Diabetes

Three comparisons of diseased to healthy groups were performed: (1) 139 non-diabetic and 23 diabetic obese adults with carotid atherosclerosis from the 300-OB cohort, (2) 126 non-diabetic and 14 diabetic obese adults without atherosclerosis from the 300-OB cohort, and (3) 130 non-diabetic and 140 diabetic obese adults with coronary atherosclerosis from the 'coronary-atherosclerosis cohort'. The coronary-atherosclerosis cohort was collected as part of the HORIZON 2020 European Research Program – 'REPROGRAM: Targeting epigenetic REPROGRamming of innate immune cells in Atherosclerosis Management and other chronic inflammatory diseases'. The coronary-atherosclerosis cohort consisted of 229 Romanian patients with symptomatic stable coronary artery disease. All patients had angiographically documented coronary atherosclerosis and inducible ischemia at the treadmill or imaging stress testing. The distribution of age, gender, and BMI in this cohort is similar to the 300-OB cohort. We used diabetes cohort no. 1 as our primary analysis (*Figure 5*), whereas the two other cohorts were used as additional support (*Figure 5—figure supplement 1C* and *Figure 5—*

*figure supplement 2B*). For all three comparisons, we used the map of parameters that was originally constructed using healthy obese individuals, as demonstrated in *Figure 1A*.

## Atherosclerosis

Two cohorts were tested. (1) An atherosclerosis male cohort, consisting of 95 men with diagnosed carotid atherosclerosis (with ultrasound) and 72 healthy men as controls. All individuals are obese older adults without cerebrovascular events from the 300-OB cohort. As a reference map of parameters, we used the map of parameters that was originally constructed using obese healthy males from *Figure 1C*. (2) An atherosclerosis female cohort, consisting of 67 women with diagnosed carotid atherosclerosis (with ultrasound) and 68 healthy women as controls. All individuals are obese older adults without cerebrovascular events from the 300-OB cohort. As a reference map of parameters, we used the map of parameters that was originally constructed using obese healthy females from *Figure 1C*. The two cohorts refer to asymptomatic atherosclerosis.

*Supplementary file 2E* provides details about the groups of disease and control subjects and about particular map of parameters that was used in each analysis.

## Application of the framework in disease research

We used the framework to compare healthy and diseased groups that have matching characteristics of age, gender, and BMI. All comparisons relied on a standard score of association between disease and a certain parameter. The 'conventional association' score was a –log p-value resulting from a linear mixed model that accounted for relevant covariates of gender, age, and BMI. We used the same formulation to test the association between each of the IM-state scores and disease. In addition, we also calculated associations between a given clinical parameter (or a gene parameter) and disease while controlling for the whole-body state – that is, we used both the IM1-state and IM2-state scores as additional covariates. The –log p-value resulting from this test is referred to as a 'state-independent association' score. Reported association p-values were adjusted for false discovery rate that accounts for multiple testing (*q*-values).

For comparison of subjects with disease to healthy subjects, the map of parameters should be constructed using matching healthy individuals from an independent cohort. For instance, we applied this strategy in the case of SLE, where the map of parameter was constructed using one cohort (the normal-BMI cohort) and the compared healthy/disease groups were derived from another cohort (*Supplementary file 2E*). Given that a matching healthy cohort is not always available for the construction of a map (e.g., the available independent cohorts do not have a similar list of parameters), the same healthy controls could be used for the map construction and for the healthy/disease comparison. In such case, to avoid overfitting and biases, the IM-state scores of healthy individuals are inferred in a leave-one-out manner – namely, constructing the map multiple times, each time omitting data from one particular individual for which the scores are inferred. We applied this strategy in the case of atherosclerosis and diabetes, where both the reference map and the control groups relied on the same cohort (*Supplementary file 2E*).

We reasoned that clinical parameters could be altered in disease independently of the state and that these parameters could have an effect on the calculation of the state in disease patients. To decouple between the state and disease-specific alterations, we refined the calculated IM1/IM2-state scores of disease patients in an iterative manner. In each iteration, the procedure removed parameters that were significantly associated independently of the two state scores and then recalculated the states. These two steps were iterated until convergence. To ensure comparability, the final set of parameters that were used to infer the two state scores in disease patients were also used to infer the two scores in the group of healthy individuals. None of the parameters were omitted in analysis of atherosclerosis. The set of significant state-independent SLE associations (*Supplementary file 2G*) was omitted in SLE. Glucose and Hb1ac were omitted in diabetes.

## Acknowledgements

We thank all of the volunteers in the 300-OB cohort for their participation. This work was supported by the Israel Science Foundation (ISF) grant 288/16, by a European Research Council grant (ERC 637885), and by IN-CONTROL CVON grants (CVON2012-03 and CVON2018-27) of the Netherlands Heart Foundation. LABJ is supported by a Competitiveness Operational Programme grant of the

Romanian Ministry of European Funds (HINT, ID P_37_762; MySMIS 103587). IG-V is a Faculty Fellow of the Edmond J Safra Center for Bioinformatics at Tel Aviv University, and is supported and by the European Union Horizon 2020 under grant agreement No. 847422. AF was supported by ISF288/16, by the Edmond J Safra Center for Bioinformatics at Tel Aviv University, and by ERC 637885. MGN is supported by a Netherlands Organization for Scientific Research Spinoza Grant (NWO SPI 94-212) and a European Research Council Advanced grant (ERC 833247 to MGN). NPR, LABJ, MGN, ACI, IMD and BAT received funding from the European Union Horizon 2020 research and innovation program REPROGRAM under grant agreement No 667837. NPR received a JTC2018 grant ('MEMORY') from the European Research Area Network on Cardiovascular Disease (ERA-CVD).

## Additional information

### Funding

| Funder | Grant reference number | Author |
|---|---|---|
| European Commission | 637885 | Amit Frishberg<br>Irit Gat-Viks |
| Israel Science Foundation | 288/16 | Amit Frishberg<br>Irit Gat-Viks |
| Hartstichting | IN-CONTROL CVON2012-03 | Niels P Riksen<br>Mihai G Netea |
| Hartstichting | IN-CONTROL CVON2018-27 | Niels P Riksen<br>Mihai G Netea |
| Romanian Ministry of European Funds | HINT, ID P_37_762; MySMIS 103587 | Leo AB Joosten |
| Horizon 2020 | 847422 | Irit Gat-Viks |
| Tel Aviv University | Edmond J. Safra Center for Bioinformatics | Amit Frishberg<br>Irit Gat-Viks |
| Netherlands Organisation for Scientific Research | Spinoza Grant NWO SPI 94-212 | Niels P Riksen<br>Mihai G Netea |
| European Research Council | 833247 | Mihai G Netea |
| Horizon 2020 | REPROGRAM 667837 | Niels P Riksen<br>Leo AB Joosten<br>Mihai G Netea<br>Adrian C Iancu<br>Ioana M Dregoesc<br>Bogdan A Tigu |
| Horizon 2020 | European Research Area Network on Cardiovascular Disease JTC2018 'MEMORY' | Niels P Riksen |

The funders had no role in study design, data collection and interpretation, or the decision to submit the work for publication.

### Author contributions

Amit Frishberg, Conceptualization, Data curation, Software, Formal analysis, Validation, Investigation, Visualization, Methodology, Writing - original draft; Inge van den Munckhof, Resources, Data curation, Formal analysis, Methodology, Writing - original draft; Rob ter Horst, Resources, Data curation, Formal analysis; Kiki Schraa, Data curation, Formal analysis, Methodology; Leo AB Joosten, Conceptualization, Data curation, Funding acquisition, Investigation, Methodology; Joost HW Rutten, Conceptualization, Resources, Data curation, Formal analysis, Methodology; Adrian C Iancu, Resources, Writing - original draft; Ioana M Dregoesc, Resources, Data curation, Methodology; Bogdan A Tigu, Resources, Data curation, Writing - original draft; Mihai G Netea, Conceptualization, Resources, Formal analysis, Validation, Investigation, Methodology, Writing - original draft; Niels P Riksen, Conceptualization, Resources, Data curation, Supervision, Validation, Investigation,

Methodology, Writing - original draft; Irit Gat-Viks, Conceptualization, Supervision, Investigation, Methodology, Writing - original draft, Project administration

### Author ORCIDs
Amit Frishberg ⬵ https://orcid.org/0000-0002-6912-9801
Leo AB Joosten ⬵ http://orcid.org/0000-0001-6166-9830
Ioana M Dregoesc ⬵ https://orcid.org/0000-0003-0340-4476
Bogdan A Tigu ⬵ http://orcid.org/0000-0001-9397-0791
Niels P Riksen ⬵ https://orcid.org/0000-0001-9197-8124
Irit Gat-Viks ⬵ https://orcid.org/0000-0002-5431-6444

### Decision letter and Author response
Decision letter https://doi.org/10.7554/eLife.61710.sa1
Author response https://doi.org/10.7554/eLife.61710.sa2

## Additional files

### Supplementary files
• Supplementary file 1. Individuals and clinical parameters. (A) Individuals in the 300-OB cohort. For each individual (column 1), the table reports the gender, age, and body mass index (BMI) (columns 2–4), the presence of diabetes mellitus (column 5), and carotid atherosclerosis (column 6). Indications for atherosclerosis (maximum plaque thickness, maximum stenosis, and number of plaques) are in columns 7–9, respectively. The healthy controls (no atherosclerosis and no diabetes) are referred to as the healthy 'obesity cohort'. (B) Individuals in the normal BMI cohort. The table reports the index of individuals included in the normal BMI cohort (column 1). Gender, age, and BMI of each individual are reported in columns 2–4, respectively. (C) Clinical parameters. The table reports the list of parameters measured in the 300-OB cohort and/or the normal-BMI cohort that were included in this study (the obesity cohort is part of the 300-OB cohort). For each parameter (column 1), indicated is the physiological system (column 2) and whether the parameter was measured in the 300-OB cohort (column 3) and normal-BMI cohort (column 4). (D) Immune capacity and lipoprotein parameters used in this study. The table provides a summary for the immune capacity and lipoprotein parameters in the 300-OB cohort and/or the normal-BMI cohort. Each of these parameters is listed with more details in (C).

• Supplementary file 2. Analysis of biological functions and disease, performed in the context of the map and the whole-body state. (A) The parameters. For each parameter (column 1), columns 2 and 3 report the position of parameter in the map, as calculated using the obesity cohort data (IM1 and IM2 axes, respectively), and columns 4 and 5 report the position of parameter in the map as calculated using the normal BMI cohort data (IM1 and IM2 axes, respectively). Columns 6 and 7 report the Pearson's correlations (r) between each parameter and the state, as calculated using the obesity cohort data (IM1 and IM2 states, respectively), and columns 8 and 9 provide the correlation (r) between each parameter and the state, as calculated using the normal-BMI cohort data (IM1 and IM2 states, respectively). (B) Pearson correlations between each pair of parameters. Parameters are ordered as in *Figure 1—figure supplement 3D*. Correlations were calculated using the covariate-corrected transformed data of the obesity cohort. (C) Top state-specific markers. Reported are top identified markers (columns 1 and 2) together with their correlation with the IM1 state (columns 3 and 5) and IM2 state (columns 4 and 6) in the obesity cohort (columns 3 and 4) and normal-BMI cohort (columns 5 and 6). Column 7 reports the association with metabolic syndrome (MetS). (D) Associations of functional categories with the whole-body state. For each functional annotation (columns 1 and 2), the table reports the significance of association with the IM1 and IM2 states (columns 3 and 4, respectively). Reported are (signed) q-values of a function-state association test. Positive/ negative values are indicative of up/downregulation of IM1-positive (or IM2-positive) individuals. Included are significant functional annotations based on the extended map from *Figure 3—figure supplement 2B*. (E) Comparisons between health and disease groups in this study. For each comparison (column 1), reported is the parameter map used as a reference (columns 2 and 3), and the compared healthy and disease groups (columns 4–8). (F) The parameter map and associations with

atherosclerosis and CVD. For each parameter (column 1), column 2 reports the conventional scores of association with atherosclerosis, and column 3 reports the state-independent association with atherosclerosis (–log p-values, with positive values indicative of higher values in atherosclerosis subjects compared to those without and negative values indicative of lower values in the atherosclerosis subjects). Previous findings about CVD risk factors are reported in columns 4 and 5. Reported are CVD risk factors in at least one age/BMI/gender/ethnicity group and excluding controversial factors. (G) State-independent associations with systemic lupus erythematosus (SLE). Reported are genes that are significantly associated with SLE independently of the state and their state-independent association scores (–log q-value). Association scores are signed according to the direction of associations; for instance, 76 genes are positively associated and 12 genes are negatively associated with SLE in female children. Results are reported for the analysis of three data sets (E). Known interferon signaling genes are highlighted in bold. *For SLE in males, due to the small number of individuals in this cohort, reported are –log p-values.

- Transparent reporting form

### Data availability

Both the obesity cohort and the normal BMI cohort were part of the Human Functional Genomics Project (http://www.humanfunctionalgenomics.org/site/) and have been previously published. The coronary-atherosclerosis cohort was collected as part of the HORIZON 2020 European Research Program - "REPROGRAM: Targeting epigenetic REPROGRamming of innate immune cells in Atherosclerosis Management and other chronic inflammatory diseases". SLE sequencing public datasets used in our analysis: GSE65391, GSE49454.

The following previously published datasets were used:

| Author(s) | Year | Dataset title | Dataset URL | Database and Identifier |
|---|---|---|---|---|
| Banchereau R, Hong S, Cantarel B, Baldwin N, Baisch J, Edens M, Cepika A, Acs P, Turner J, Anguiano E, Vinod P, Kahn S, Obermoser G, Blankenship D, Wakeland E, Nassi L, Gotte A, Punaro M, Liu Y, Banchereau J, Rossello-Urgell J, Wright T, Pascual V | 2016 | Longitudinal transcriptional pediatric SLE study with clinical parameters | https://www.ncbi.nlm.nih.gov/geo/query/acc.cgi?acc=GSE65391 | NCBI Gene Expression Omnibus, GSE65391 |
| Chiche L, Jourde-Chiche N | 2014 | Modular repertoire analyses identify dynamic type I and type II interferon transcriptional signatures in adult SLE patients | https://www.ncbi.nlm.nih.gov/geo/query/acc.cgi?acc=GSE49454 | NCBI Gene Expression Omnibus, GSE49454 |

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
