## [Decision Letter]

**Acceptance summary:**

Human diseases arise in a complex ecosystem composed of disease mechanisms and the whole-body state. The authors have provided novel insights having mapped a large collection of metabolic, hemodynamic and immune parameters, used the mapping to dissect phenotypic states and found that the whole-body state is faithfully represented by a quantitative two-dimensional model. Both components are associated with disease, but differ in their particular associations, thus opening new avenues for improved personalized diagnosis and treatment.

**Decision letter after peer review:**

Thank you for submitting your article "An integrative model of cardiometabolic traits identifies two types of metabolic syndrome" for consideration by *eLife*. Your article has been reviewed by three peer reviewers, including Edward D Janus as the Reviewing Editor and Reviewer #1, and the evaluation has been overseen by a Reviewing Editor and a Senior Editor. The following individual involved in review of your submission has agreed to reveal their identity: Chao Wang (Reviewer #1).

The reviewers have discussed the reviews with one another and the Reviewing Editor has drafted this decision to help you prepare a revised submission.

As the editors have judged that your manuscript is of interest, but as described below revisions are required before it is published, we would like to draw your attention to changes in our revision policy that we have made in response to COVID-19 (https://elifesciences.org/articles/57162). First, because many researchers have temporarily lost access to the labs, we will give authors as much time as they need to submit revised manuscripts. We are also offering, if you choose, to post the manuscript to bioRxiv (if it is not already there) along with this decision letter and a formal designation that the manuscript is "in revision at *eLife*". Please let us know if you would like to pursue this option. (If your work is more suitable for medRxiv, you will need to post the preprint yourself, as the mechanisms for us to do so are still in development.)

Summary:

Irit Gat-Viks and coauthors attempted to address an intriguing and ambitious question: what constitutes an individual's whole body state. It is an exciting study but lacks some details. It expands the established concept of metabolic syndrome predisposing to diabetes and cardiovascular disease. From very detailed analysis of many metabolic, immunological and haemodynamic parameters it defines two types of metabolic syndrome IM 1 largely representing the already established metabolic syndrome and a new IM 2 non classical metabolic syndrome. Disease states are associated with either or both.

Overall, the study is very exciting and provides a thought-provoking challenge and potentially an important new paradigm. Providing further specific details as requested below would benefit general readers. The manuscript is however very long and difficult to follow in parts, so it needs to be simplified. Try to streamline the presentation and abbreviate the findings and main message to have impact.

Essential revisions:

(1) Style and overall issues: The manuscript especially the Results and Discussion should be shortened considerably and the key points need to be stated very clearly and explicitly as in this draft the messages get lost in the detail. The objective of each section needs to be set out clearly at the outset – as it already is for some – and the conclusion from each section needs to be stated clearly and concisely at the end of each section.

The final 8 parameters of interest are in supplementary figure 8 D but get lost there.

The reader at the outset expects that some of the immunological parameters may feature in the second type IM – 2 but in the end both types are primarily based on lipid parameters. This should be explained in the discussion.

The Abstract does not give information on the statistical methods employed.

The Discussion should start with the main findings. The Discussion should end with a clear conclusion and clarify how the findings can be utilized in future research and or clinically.

2) What drives the IM1 and IM2 state in their PCA plot for each of the cohort the authors investigated? The authors refer to 8 measured parameters being "sufficient" in characterizing the whole body state, and they are all lipids. How were they chosen? What would be informative is what other types of parameters drives the IM1/2 state in normal BMI and obesity cohort especially in the map of parameters where lipid transformation is used.

2) The authors subsequently generated an extended map with RNAseq data from normal BMI cohort. How gene expression data correlate with their clinical/non-clinical parameters which they initially used to define the whole body state? In particular, what are the top genes/pathways associated with IM1/2 state in healthy BMI cohort? Detailed heatmap or other comprehensive visualization should be used rather than hand picking a few pathway such as the IFN response pathway. This should be done before characterizing disease state. Overall, the study gave us an exciting proposition of a whole body state but does not tell us what constitutes the whole body state (namely what drives IM1/2).

3) Unbundling of the MetS has been attempted using cluster analysis and a principal components approach. So, what does this new approach add? How reproducible are the findings? What are the consequences of this new knowledge? What are the practical implications of the investigation? What is the value of knowing the 2 subtypes identified? More discussion required.

4) In Figure 1A, it is important to indicate on the map (Figure 1A) where each type of clinical/non-clinical parameters are located (color code perhaps such as in Figure 1—figure supplement 5). While we appreciate it is not possible to highlight every single parameter, it would be of interest to readers in general to visualize covariance of parameters within the same category. For example, it is intriguing to see IL-6 on the opposite spectrum of IL-1b/TNFa, which is not necessarily expected. The authors measured IL-6 in a wide variety of conditions. Which IL-6 did this come from? The authors spent a lot of effort demonstrating the power of their data transformation looking at single parameter correlates, but it would be of more benefit for the readers to appreciate their results in a comprehensive manner: can they visualize the correlation coefficient or IM1/2 coordinates in the form of a heatmap for example rather than a supplementary table?

5) In Figure 1D, what does the author mean by "The two maps (normal BMI and obesity) are very similar in their general architecture"? Do the authors refer to the general position of the same clinical/non-clinical parameter on the map? What happens to parameters that are not measured in both cohort? The authors state the normal BMI and obesity cohort has different parameter measured. For example, based on Supplementary file 1D, completely different immune/cytokine parameters are measured for each cohort. Are immune/cytokine parameter not at all in the map in Figure 1D? If they are, where do they land? Again, a heatmap of all parameters would be beneficial for also the normal BMI cohort.

6) It is unclear how the authors picked the genes (only 596 of 7443 available) from SLE cohort for example to generate the map as they didn't have clinical parameters measured. Did the authors picked genes correlated with lipid enzymes?

[Editors' note: further revisions were suggested prior to acceptance, as described below.]

Thank you for choosing to send your work entitled "An integrative model of cardiometabolic traits identifies two types of metabolic syndrome" for consideration at *eLife*. Your resubmission and responses have been assessed by a Senior Editor in consultation with a member of the Board of Reviewing Editors.

The authors need to (Q1) highlight and discuss the 8 key parameters and (Q2) relate these to what is currently referred to the "metabolic syndrome". (Q3) The authors will thereby provide a perspective on their concepts for the reader.

---

## [Author Response]

Essential revisions:(1a) Style and overall issues: The manuscript especially the Results and Discussion should be shortened considerably and the key points need to be stated very clearly and explicitly as in this draft the messages get lost in the detail. The objective of each section needs to be set out clearly at the outset – as it already is for some – and the conclusion from each section needs to be stated clearly and concisely at the end of each section.

As suggested, we substantially revised the Results by moving details to Materials and methods, adding conclusions at the end of each section and refining the text. We also shortened the Discussion section with a focus on the main points. Due to this reorganization, some of the Figures that originally appeared in Supplementary figure 3, Supplementary figure 4 are now included in Figure 1—figure supplement 6.

(1b) The final 8 parameters of interest are in supplementary figure 8 D but get lost there.

We thank the reviewer for this comment. We now focus on the eight parameters in the revised Figure 2D. We also add additional demonstration for the eight parameters in the new Figure 2D-II.

(1c) The reader at the outset expects that some of the immunological parameters may feature in the second type IM – 2 but in the end both types are primarily based on lipid parameters. This should be explained in the discussion.

Two observations explain the particular selection of eight parameters. First, the best markers for the states reside at the extreme sides of each axis, implying that the markers should be selected from these regions. Second, each category of parameters (immune, hemodynamic, metabolism, lipoproteins) have a different distribution over the map, and the relevant regions at the extremities of the axes primarily contain metabolism/lipoproteins. Thus, any eight selected parameters from these regions would be based on metabolic/lipoprotein parameters.

We now added several new visualizations to emphasize these points: (1) The organization of parameter-to-state correlations over the map (new Figure 2—figure supplement 3A), and the distribution over the map of the variance that is explained by IM1 and IM2 (new Figure 2—figure supplement 3B); together, these visualizations indicate that the markers should be chosen from the extreme sides of each axis. (2) The distribution of different categories of parameters over the map (new Figure 1D, new Figure 1—figure supplement 4B), indicating the lack of immune/hemodynamic parameters in the relevant regions of marker selection. We now highlight these points in the Results section.

(1d) The Abstract does not give information on the statistical methods employed.

We now highlight the pipeline of the methodology (building a map and then deriving a state based on the map) in the Abstract.

(1e) The Discussion should start with the main findings. The Discussion should end with a clear conclusion and clarify how the findings can be utilized in future research and or clinically.

We agree with the reviewer and apologize for the lack of clarity in the Discussion. As suggested, we have shortened the Discussion and now focus only on the main findings (first and third paragraphs), and three main applications (fourth to fifth paragraphs).

2) What drives the IM1 and IM2 state in their PCA plot for each of the cohort the authors investigated? The authors refer to 8 measured parameters being "sufficient" in characterizing the whole body state, and they are all lipids. How were they chosen? What would be informative is what other types of parameters drives the IM1/2 state in normal BMI and obesity cohort especially in the map of parameters where lipid transformation is used.

We agree with the reviewers that our original manuscript did not present sufficient information on the relations between parameters and the whole-body state, nor did we provide sufficient explanation for the selection of the eight markers. In the revised manuscript we therefore made the following changes:

1) We added two systematic analyses that together indicate the relation of each parameter with the wholebody state: (i) For each parameter, analysis of parameter-to-state correlation, depicted on top of the map (new Figure 2—figure supplement 3A); and (ii) For each parameter, the fraction of its total variation that is explained by the state (repeating these analyses for the IM1 state, IM2 state, or both) (new Figure 2—figure supplement 3B). These analyses indicate that all parameters residing nearby the extreme sides of each axis are tightly related to the corresponding state (subsection “The map enables identification of two main components of the whole-body state – the IM1 and IM2 states”).

2) We clearly define how the markers were selected (subsection “The map enables identification of two main components of the whole-body state – the IM1 and IM2 states”). In particular, as the two analyses above indicate that the markers should be selected from the extremities of each axis, the set of eight markers consists of the two most extreme parameters from each side of each axis. We further directly demonstrate the relations between these selected markers and the state (new Figure 2D-II).

3) We show that different selections of 8 markers provide similar accuracy, as long as the 8 markers are chosen from the two extreme sides of each axis (new Figure 2—figure supplement 3D).

4) We provide the distribution of each type of parameters (new Figure 1D). The fact that the 8 selected parameters are mainly lipids is in agreement with these distributions.

5) We show that the eight selected markers, which were originally selected based on the obesity cohort, perform well in the normal-BMI cohort (new Figure 2—figure supplement 3C).

6) We highlight these five points in the text (Results). We believe that these changes cover the various concerns that were raised by the reviewers.

3) The authors subsequently generated an extended map with RNAseq data from normal BMI cohort. How gene expression data correlate with their clinical/non-clinical parameters which they initially used to define the whole body state? In particular, what are the top genes/pathways associated with IM1/2 state in healthy BMI cohort? Detailed heatmap or other comprehensive visualization should be used rather than hand picking a few pathway such as the IFN response pathway. This should be done before characterizing disease state. Overall, the study gave us an exciting proposition of a whole body state but does not tell us what constitutes the whole body state (namely what drives IM1/2).

The reviewers raised two concerns, one related to the relations of genes with the IM1/2 states, and the other one related to the relations of biological functions with the IM1/IM2 states.

1) Relations of IM1/2 states with genes. We now added an analysis of the relations between each gene and the state (either IM1, IM2, or both) (new Figure 2—figure supplement 3A).

2) Relations of IM1/2 states with biological functions. The original manuscript included systematic analysis of functional annotations and their relations with the IM1/IM2 states. Following the reviewer’s comment, we realized that the source of this confusion was the fact that these results were presented only in supplementary table (i.e., not in the Results section, nor in any visualization). We therefore made the following changes. First, we added an overview presentation for the association of each functional annotation with the IM1 and IM2 axes (new Figure 1—figure supplement 5B). Several of the functions are highlighted in this new visualization (revised Figure 1—figure supplement 5C). Second, we provide more details about the organization of biological functions over the map (Results) and the implied relations between functions and the IM1/2 states (subsection “The map enables identification of two main components of the whole-body state – the IM1 and IM2 states”). The functional characterization of the map (and the states) is reported before the analysis of disease, as suggested by the reviewers. For clarity, we have changed the name of the relevant statistical test (now “areal pattern test”, formerly the “spatial colocalization test”). We improved the description of the method accordingly (subsection “The map enables identification of two main components of the whole-body state – the IM1 and IM2 states”).

4) Unbundling of the MetS has been attempted using cluster analysis and a principal components approach. So, what does this new approach add? How reproducible are the findings? What are the consequences of this new knowledge? What are the practical implications of the investigation? What is the value of knowing the 2 subtypes identified? More discussion required.

We thank the reviewer for this suggestion. We now revised substantially the Discussion section to highlight these points. Specifically, the IM1 state coincides with the known MetS categorization, and the IM2 state is suggested as a novel decoupled type of metabolic syndrome (second paragraph); reproducibility is emphasized (first paragraph), and the implications for disease research is demonstrated and discussed.

5) In Figure 1A, it is important to indicate on the map (Figure 1A) where each type of clinical/non-clinical parameters are located (color code perhaps such as in Figure 1—figure supplement 5). While we appreciate it is not possible to highlight every single parameter, it would be of interest to readers in general to visualize covariance of parameters within the same category. For example, it is intriguing to see IL-6 on the opposite spectrum of IL-1b/TNFa, which is not necessarily expected. The authors measured IL-6 in a wide variety of conditions. Which IL-6 did this come from? The authors spent a lot of effort demonstrating the power of their data transformation looking at single parameter correlates, but it would be of more benefit for the readers to appreciate their results in a comprehensive manner: can they visualize the correlation coefficient or IM1/2 coordinates in the form of a heatmap for example rather than a supplementary table?

We apologize for the lack of clarity, and have revised the text and figures to address these questions. (1) We now provide a comprehensive visualization (a heatmap) of correlation coefficients between all pairs of parameters (new Figure 1—figure supplement 3D, Figure 1—figure supplement 4A). We also provide the same information in a textual format (new Supplementary file 2B) and revised the text accordingly (Results). As expected, this visualization highlights the global structure of correlations between nearby parameters and anti-correlations between parameters in opposite sides of the map. (2) In the original manuscript we referred to the total concentration of IL6 in blood as “IL6”, whereas the ex vivo capacity of isolated PBMCs of IL6 secretion in response to a particular pathogen is referred to as “IL6/pathogen-name”. We agree that this is confusing. To address this, we now use the term “circulating IL6” instead of “IL6” (e.g., Supplementary file 2A, revised Figure 1A, new Figure 1—figure supplement 3C) and the text is now explicit on this point (Materials and methods [legends of new Supplementary figure 3C]). (**3**) As suggested, we added examples for the unexpected anticorrelation between the concentration of IL6 in circulation and the secretion capacity of different cytokines (IL-6/IL-1β/TNF-α) in response to PHA stimulation (new Figure 1—figure supplement 3C).

6) In Figure 1D, what does the author mean by "The two maps (normal BMI and obesity) are very similar in their general architecture"? Do the authors refer to the general position of the same clinical/non-clinical parameter on the map? What happens to parameters that are not measured in both cohort? The authors state the normal BMI and obesity cohort has different parameter measured. For example, based on supplementary Table 1D, completely different immune/cytokine parameters are measured for each cohort. Are immune/cytokine parameter not at all in the map in Figure 1D? If they are, where do they land? Again, a heatmap of all parameters would be beneficial for also the normal BMI cohort.

We agree with the reviewer, and apologize for the missing details in our description and supplementary information. We made the following changes: (1) As noted by the reviewers, we indeed referred to similar positions of the same parameters in the two maps. The text is now explicit on this point (Results). (2) We added two revised tables. In Supplementary file 1C, the table lists the entire collection of parameters, indicating for each parameter whether it was measured in the obesity cohort, the normal-BMI cohort, or both. In Supplementary file 2A, for each cohort, the table provides the map positions of all measured parameters (including parameters that appear in only one cohort). (**3**) We added two new visualizations that indicate the similar organization of the normal-BMI and obesity maps: first, the distribution of biological categories over the map (new Figure 1D versus new Figure 1—figure supplement 4B), and second, the correlation coefficients between pairs of parameters (new Figure 1—figure supplement 3D and Figure 1—figure supplement 4A). In both demonstrations, the entire set of normal-BMI parameters is presented (rather than only parameters that were measured in both cohorts).

7) It is unclear how the authors picked the genes (only 596 of 7443 available) from SLE cohort for example to generate the map as they didn't have clinical parameters measured. Did the authors picked genes correlated with lipid enzymes?

The source of confusion was our lack of clarity in the methodological description of the selection rules in the original manuscript. We did not select the genes based on data in the SLE cohort. Rather, we selected the genes based on data in the normal-BMI cohort, in which both gene expression and lipoproteins were measured. The extended map that we use for the analysis of SLE data is also the map that was originally constructed based on the normal BMI data. We have revised the text to clarify these points (Materials and methods).

[Editors' note: further revisions were suggested prior to acceptance, as described below.]The authors need to (Q1) highlight and discuss the 8 key parameters and (Q2) relate these to what is currently referred to the "metabolic syndrome". (Q3) The authors will thereby provide a perspective on their concepts for the reader.

The reviewers pointed out the lack of a comprehensive analysis of markers, which we apologize for and now address in full. We completely agree that markers could be used to characterize the states. As the reviewers point out, this may provide an important perspective on the concept and biological meaning of states and on the particular relations with MetS. Following the constructive comments raised, we revised our manuscript and included additional visualizations.

(Q1) highlight and discuss the 8 key parameters.

We added a thorough and unbiased analysis of the top IM1-specific and IM2-specific markers. Specifically, as we originally introduced only 8 markers that were selected arbitrarily from a larger set of 24 most-accurate markers, the list of top markers is now extended to include the entire set of best 24 markers. We include several alternative visualizations of these top markers (new Figure 3DE, new Figure 3—figure supplement 1A, new Supplementary file 2C), demonstrate the top markers in the larger context of the lipoprotein pathways (new Figure 3F, new Figure 3—figure supplement 1B-E), discuss potential biological mechanisms that could underlie the observed signatures (Results, new Figure 3—figure supplement 1F), highlight the top markers (Discussion section and Abstract), and provide the precise method for the selection of these markers (Materials and methods). In addition, we also include demonstration of significant markers that are not the top-significant markers (new Figure 3A-C, Results). Thus, the overall characterization of the state relies on insights from top significant markers, additional significant markers, and functional annotations (Results, Discussion). We thank the reviewer for this suggestion, which we believe strengthened our work.

We now realized that the difference between single markers for the state and state-prediction using a combination of markers is not clear. We originally used the combination of 8 markers to approximate the prediction of the state. These 8 markers were selected arbitrarily from a larger set of 24 best candidate markers. While we did not intend to emphasize any particular combination, it was obviously highly confusing. In the revised manuscript, we clarify the difference between these two analyses. First, we moved the two analyses to two separate sections (single markers: Results; marker combinations: Discussion). In accordance, prediction accuracy was moved to Figure 3—figure supplement 3 (former Figure 2D). Second, we added a separate methodological explanation for each of the analyses (single markers: ; marker-combinations: Materials and methods). Third, we refined the description of state-prediction in the Results section to emphasize the differences from a simple single-marker analysis. We now highlight the fact that all combinations of parameters (selected from the top 24 parameters) have high accuracy (revised Results and revised Figure 3—figure supplement 3A), implying that there is no particular interest in one specific combination. Fourth, the original combination of 8 parameters (for which we show prediction accuracy in Figure 3—figure supplement 3B) is demonstrated in detail in new Figure 3E and highlighted together with the remaining top markers in the Results, thus directly addressing the reviewer’s request.

For simplicity, we now present the analysis of functional annotations as a natural extension of the analysis of markers (revised Results, Materials and methods). We note that the reported geneset *P*-values in Figure 3—figure supplement 2A and Supplementary file 2D (former Figure 1—figure supplement 5B and Supplementary file 2C) are slightly different from the original submission due to two main changes: First, we revised the way we calculate the P-values in order to present the analysis as an extension of single-marker analysis (now referred to as “function-state association”, formerly referred to as “areal pattern”). Second, we now use a larger number of permutations for the calculation of P-values. As the former and new formulation are equivalent, and indeed provide very similar P-values, there is no change in the identified functions.

(Q2) relate these to what is currently referred to the "metabolic syndrome".

The relations of the identified state-specific markers with MetS are now demonstrated in several ways. First, we directly demonstrate that the associations of parameters with the IM1 state are similar to the associations of parameters with MetS (new Figure 4D). Second, we show that the top 12 markers of IM1 are significantly associated with MetS (highlighted in new Figure 4D), whereas the top 12 markers of IM2 are not associated with MetS (new Supplementary file 2C). Third, we show how each single MetS parameter relates to the IM1 state (new Figure 4—figure supplement 1B) and how the combination of the five MetS parameters predicts the IM1 state with high accuracy (new Figure 4—figure supplement 1C). These findings provide important insights: (i) a list of alternative markers that could be useful in MetS diagnosis, potentially leading to a stronger association of classical MetS with cardiometabolic disease; and (ii) novel IM1-associated parameters, the role of which in MetS is still unknown. We now clarify these points in the Results section and Discussion section.

(Q3) The authors will thereby provide a perspective on their concepts for the reader.

The revised manuscript describes more clearly the relevance of our model, which (1) provides a novel framework that can reliably describe different phenotypic states in humans and that could better predict the presence of disease, (2) provides individual markers that could potentially improve individual risk/disease prediction, (3) adds to the biological understanding of which biological processes drive disease states (Discussion). The biological understanding is obtained based on various new analyses, including analysis of individual markers (new Figure 3A-C, Results), top-significant markers (new Figure 3D-F, Results), and highlighting additional examples of functional annotations (Results).